



# Concurrent variation in oil and gas methane emissions and oil price during the COVID-19 pandemic

David R. Lyon[1*], Benjamin Hmiel[1], Ritesh Gautam[1], Mark Omara[1], Kate Roberts[1], Zachary R. Barkley[2], Kenneth J. Davis[2], Natasha L. Miles[2], Vanessa C. Monteiro[2], Scott J. Richardson[2], Stephen Conley[3], Mackenzie L. Smith[3], Daniel J. Jacob[4], Lu Shen[4], Daniel J. Varon[4], Aijun Deng[5], Xander Rudelis[6, †], Nikhil Sharma[6], Kyle T. Story[6], Adam R. Brandt[7], Mary Kang[8], Eric A. Kort[9], Anthony J. Marchese[10], Steven P. Hamburg[1]

[1]Environmental Defense Fund, 301 Congress Ave., Suite 1300, Austin, TX, USA.

[2]The Pennsylvania State University, University Park, PA, USA.

[3]Scientific Aviation, Boulder, CO, USA.

[4]Harvard University, Cambridge, MA, USA.

[5]Utopus Insights, Inc., Valhalla, NY, USA.

[6]Descartes Labs, Santa Fe, NM, USA.

[7]Stanford University, Palo Alto, CA, USA.

[8]McGill University, Montreal, QB, Canada.

[9]University of Michigan, Ann Arbor, MI, USA.

[10]Colorado State University, Fort Collins, CO, USA.

[†] now at Google LLC, Mountain View, CA, USA.

*Correspondence to*: David R. Lyon (dlyon@edf.org)

**Abstract.** Methane emissions associated with the production, transport, and use of oil and natural gas increase the climatic impacts of energy use; however, little is known about how emissions vary temporally and with commodity prices. We present airborne and ground-based data, supported by satellite observations, to measure weekly to monthly changes in total methane emissions in the United States' Permian Basin during a period of volatile oil prices associated with the COVID-19 pandemic. As oil prices declined from ~$60 to $20 per barrel, emissions changed concurrently from 3.4% to 1.5% of gas production; as prices partially recovered, emissions increased back to near initial values. Concurrently, total oil and natural gas production only declined by a maximum of ~10% from the peak values seen in the months prior to the crash. Activity data indicate that a rapid decline in well development and subsequent effects on associated gas flaring and midstream infrastructure throughput are the likely drivers of temporary emission reductions. Our results, along with past satellite observations, suggest that under more typical price conditions, the Permian Basin is in a state of overcapacity in which rapidly growing natural gas production exceeds midstream capacity and leads to high methane emissions.



## 1 Introduction

Accurate quantification of methane ($CH_4$) emissions from the oil and natural gas (O&G) supply chain is critical for determining the climatic impact of O&G production and use (Alvarez et al., 2012). Alvarez et al. (2018) synthesized over 400 site- and basin-level measurements to estimate United States O&G supply chain emissions at 13 Tg $CH_4$ in 2015, equivalent to 2.3% of the nation's natural gas production and over 80% higher than the U.S. Environmental Protection Agency (USEPA)'s bottom-up estimate (USEPA, 2020a). There is growing evidence of systematic underestimation of O&G methane emissions when bottom-up methods such as emission factors and engineering equations are used rather than top-down, atmospheric measurements (Allen, 2014;Brandt et al., 2014;Zavala-Araiza et al., 2017). The Permian Basin (Fig. 1) is the largest oil producing basin in the U.S. and rivals the Ghawar Field in Saudi Arabia for the global record (Jacobs, 2019). Although the first oil well was drilled in the Permian Basin nearly 100 years ago, the basin has experienced rapid growth in recent years as directional drilling and hydraulic fracturing allowed production from unconventional reservoirs (Enverus, 2020). In 2019, the Permian Basin had ~600 new wells drilled per month and produced an average of 4.3 million barrels (bbl) oil and 15 billion cubic feet (Bcf) natural gas per day, more than double the 2016 average values (Enverus, 2020). The Permian Basin's limited midstream gathering and processing (G&P) infrastructure for delivering natural gas to market results in high rates of associated gas flaring relative to other U.S. basins. In 2019, average daily flared gas volumes were 0.8 Bcf, 5% of the basin's natural gas production (Appendix A). There is limited methane emissions data from the Permian beyond two recent studies (Zhang et al., 2020;Robertson et al., 2020). Zhang et al. (2020) used satellite observations from May 2018 – March 2019 in an atmospheric inversion to estimate total O&G related emissions in the Permian Basin of 2.7 Tg $CH_4$ annually, or 3.7% of regional gas production. Robertson et al. (2020) found higher well pad $CH_4$ emission rates in the Permian Basin compared to most other U.S. basins based on over 70 site-level measurements made in 2018. Alvarez et al. (2018), which pre-dates these studies, had assumed other U.S. basins were representative of the Permian; updating their estimate with the Permian Basin loss rate from Zhang et al. (2020) results in a roughly 10% increase in the U.S. supply chain estimate to 14.2 Tg $CH_4$, or 2.5% of total gas production.

In January 2020, oil prices declined as the COVID-19 pandemic triggered a global slowdown in O&G consumption; in March, there was a rapid price drop when the oil oversupply was exacerbated by both the Organization of the Petroleum Exporting Countries (OPEC) failing to reach a deal to cut production and global oil storage capacity reaching its limit (Reed and Krauss, 2020). Spot prices for the U.S. oil benchmark, known as West Texas Intermediate-Cushing (WTI-Cushing), varied dramatically during this period; price per barrel was relatively stable at $50-60 (USD) for most of 2019, declined to $20 by late April 2020, briefly dropped below zero on April 20, then recovered to $40 by early July (USEIA, 2020b). Natural gas spot prices (Henry Hub) were less volatile during this period ($1.50-2.00 per million British Thermal Units), continuing a gradual downward trend since late 2018 (USEIA, 2020a). Lower commodity prices reduce investment in new well and infrastructure development; in the Permian Basin, the number of active drilling rigs, which had averaged over 400 from April 2019 to March 2020, dropped to approximately 300, 180, and 135 in April, May, and June 2020, respectively (Baker-Hughes, 2020) (Fig. 2).





We hypothesize that the rapid drop in oil price would be associated with a concomitant reduction in methane emissions due to lower rates of well development and a subsequent decline in O&G production. The postulated causal mechanism for this relationship is the effect of natural gas production from new wells on midstream infrastructure throughput. During periods of higher commodity prices, the rapid growth in natural gas production likely exceeds the capacity of the pipelines, compressor stations, and processing plants that deliver and process gas to market, leading to associated gas flaring and anomalous conditions that increase emissions. Such trends were observed in an earlier drilling slowdown in the Bakken, another U.S. unconventional oil formation (Enverus, 2020) (Fig. F1). However, this effect might have been countered in the Permian if lower profit margins led operators to allocate fewer resources to infrastructure maintenance and emissions mitigation, or similarly, restrictions due to COVID-19 reduced the number of field staff performing tasks such as leak detection and repair (LDAR) (Gould et al., 2020).



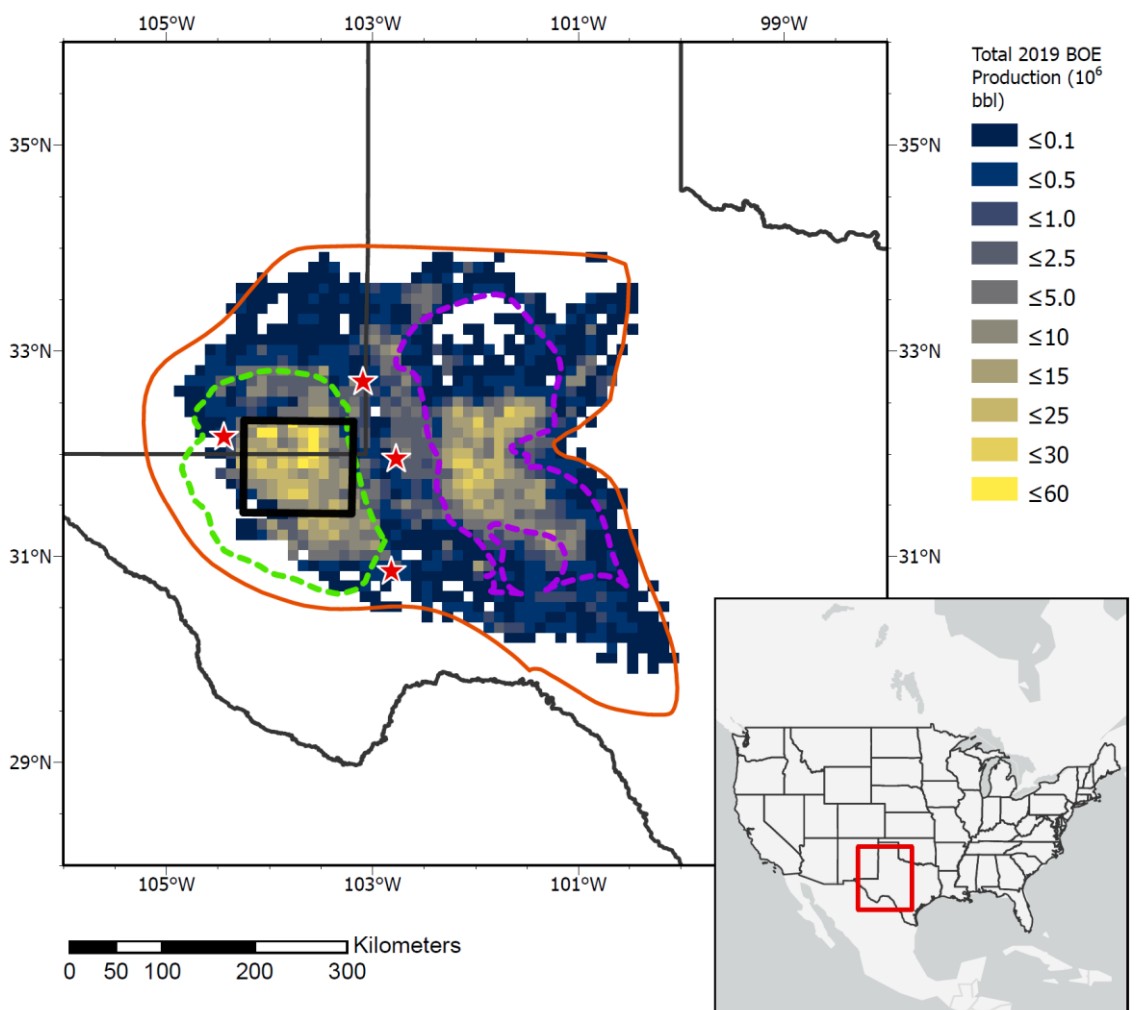

75

**Figure 1.** Regional map with outlines of the Permian Basin (orange), Delaware and Midland sub-basins (dashed green and purple) and the 100 km x 100 km study area (Black). Locations of the methane measurement tower sites are shown with red stars. A heatmap displays combined gas and oil production from 2019 expressed in barrels-of-oil equivalents (BOE) and gridded to 0.1° x 0.1° resolution (Enverus, 2020).

80



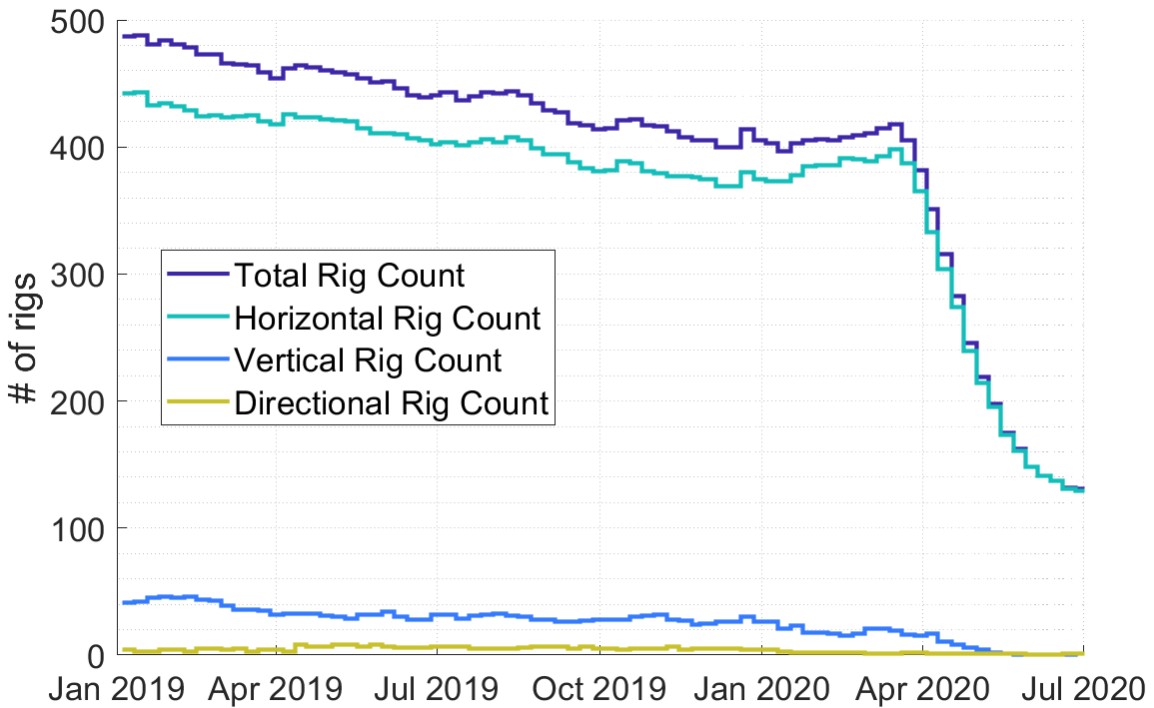

**Figure 2.** Weekly count of active drilling rigs by type in the Permian Basin between January 2019 and July 2020

85



## 2 Study Area and Methods

### 2.1 Study Area

In January 2020, we began quantifying O&G methane emissions at varying spatiotemporal scales within the Permian Basin with a concentrated effort within a 100 km x100 km area of the Delaware sub-basin along the Texas/New Mexico border (Fig. 1). The 10,000 km$^2$ study area includes ~11,000 active wells and accounts for 33% and 43% of the Permian Basin's oil and natural gas production in 2019, respectively (Enverus, 2020). The study area has a high density of midstream O&G infrastructure including 125 gathering and transmission compressor stations, 44 processing plants, and ~32,000 kilometers of gathering pipeline (Enverus, 2020). Based on spatially allocated USEPA inventory data, O&G sources accounted for >90% of methane emissions in the study area in 2012; other sources, dominated by agriculture and waste, were responsible for ~0.5 Mg $CH_4$ hr$^{-1}$ (Maasakkers et al., 2016). Since the non-O&G sources account for only a small fraction of total emissions and there have been no major changes in these activities over the past few years, we have assumed all study area emissions are attributable to O&G sources beyond the 0.5 Mg $CH_4$ hr$^{-1}$.

### 2.2 Methods

Between January and August 2020, we used two inversion approaches to quantify total methane emission flux from the study area at a weekly to monthly frequency. The first approach used aircraft-based instruments to measure atmospheric boundary layer (ABL) methane concentration ([$CH_4$]) along the study area perimeter during six daytime flights (January 22, March 9, March 25, May 4, May 21, and July 13; Sect. 2.22). The second approach continuously quantified [$CH_4$] from March through August 2020 using sensors installed at three tall towers and one mountaintop station located around the perimeter of the study area ((Richardson et al., 2017); Sect. 2.21). Both approaches estimated study area methane flux on a daily basis by optimizing a prior emissions inventory to minimize model-data differences between observed and simulated regional atmospheric [$CH_4$] ((Barkley et al., 2017); Sects. 2.21 and 2.23).

We also evaluated satellite-based remote sensing observations of column methane enhancement ($\Delta XCH_4$) for evidence of basin-wide trends (Sect. 2.14). To provide insights about the contribution of natural gas flares to methane emissions, we qualitatively assessed over 300 flares across the basin in February, March, and June 2020 using helicopter-based infrared optical gas imaging (OGI) to visually detect combustion issues ((Lyon et al., 2016); Appendix B). We estimated flare-related methane emissions by applying combustion efficiency assumptions based on survey results to flared gas volume estimates based on satellite observations of flare radiant heat ((Elvidge et al., 2016); Appendix A).

### 2.2.1 Regional atmospheric [$CH_4$] reanalysis

An atmospheric reanalysis similar to the system used in previous studies (Barkley et al., 2019;Barkley et al., 2017) was used to create simulated regional atmospheric [$CH_4$] estimates. The modeling system used Weather Research and Forecasting (WRF) model coupled with Chemistry v3.6 (Skamarock et al., 2008) configured to simulate two domains, an outer 2600 km



x 2100 km domain with 9 km x 9 km horizontal resolution and 50 vertical levels, with about 30 of these levels in the lowest 3 km above ground level, and an inner 830 km x 830 km domain with 3 km x 3 km horizontal resolution and the same vertical layers. The outer domain is nudged to ERA5 wind, temperature and water vapor reanalyses, and the inner domain is nudged

to regional observations including ~50 National Weather Service / World Meteorological Organization surface stations, five National Weather Service rawinsonde site soundings launched at 0 and 12 UT, and the meteorological measurements from commercial aircraft-ACARS. Our choice of parameterization schemes within WRF-Chem matches previous studies (Barkley et al., 2019;Barkley et al., 2017).

Only atmospheric [$CH_4$] from emissions within the model domain are simulated, using techniques demonstrated previously
(Barkley et al., 2019;Barkley et al., 2017). Preliminary estimates of surface fluxes of [$CH_4$] within the domain are taken from the EPA 2012 gridded inventory (Maasakkers et al., 2016), save for the Permian Basin where an updated, production-based inventory is used. This updated inventory is described in detail by Zhang et al. (2020). Briefly, production site $CH_4$ emission factors were developed using methods in Zavala-Araiza et al. (2015) and based on measurements by Robertson et al. (2020), which accounted for complexity of well site infrastructure and their related $CH_4$ emissions. Total basin-wide $CH_4$ emissions
were estimated using activity (Enverus, 2020) and disaggregated to individual sites based on their gas production. Additional facility-level $CH_4$ emissions for gathering and boosting stations, gathering pipelines and processing plants were estimated based on activity data (Enverus, 2020) and $CH_4$ emission factors from Marchese et al. (2015) and the EPA GHGI (USEPA, 2020a). For the transmission and storage stations, $CH_4$ emissions were taken from Maasakkers et al. (2016). For the Delaware basin, total $CH_4$ emissions were estimated at 1.2, 0.11, 0.04, and 0.01 Tg for production sites, gathering and boosting stations,
gas processing plants and gas transmission and distribution stations, respectively. These O&G $CH_4$ emissions were then spatially allocated to a $0.1° × 0.1°$ grid over the entire basin. This update within the Delaware Basin is important to account for the rapid development within the basin since 2012. Different [$CH_4$] sources (e.g. oil and gas production, landfills, agriculture) and sources inside and outside the study domain are tagged as independent tracers in the model. Oil and gas emissions outside of the study domain are multiplied by 1.6 to match estimates from Alvarez et al. (2018) and to better account
for development in the areas surrounding the study domain. This atmospheric reanalysis system enables us to create a first estimate of atmospheric [$CH_4$] consistent with the regional meteorology and the preliminary estimate of sources within the outer model domain.

Note that the emissions magnitude from the preliminary [$CH_4$] emissions estimates are not highly important since the emissions estimate is not a Bayesian inversion that assigns an uncertainty estimate to this preliminary estimate. The spatial pattern of
emissions, however, including the relative change in these spatial patterns, is important for the estimate of fluxes. Our assumption that emissions are proportional to gas production should provide a reasonable estimate of the spatial pattern of emissions corresponding to well locations.





### 2.2.2 Aircraft based methane emission estimates

The total $CH_4$ emissions from the study area in the Permian Basin study area were determined using airborne data in
conjunction with transport modeling. The airborne platform has been deployed and described previously (Conley et al., 2017;Conley et al., 2016;Karion et al., 2015;Smith et al., 2017). In brief, a single-engine Mooney aircraft is outfitted with a Picarro CRDS instrument (G2210-m) to measure in-situ atmospheric $CH_4$, $CO_2$, $H_2O$ mole fractions, a differential GPS and aircraft data computer to enable computation of horizontal wind speeds and directions, and a Vaisala probe to measure ambient temperature and relative humidity.

On each flight day, two laps consisting of a box enclosing the 100 km x 100 km study area were flown at 1100 ±100 ft above ground level (agl), with one complete lap taking ~ 2 h to complete. Two to three vertical profiles were also flown by the aircraft as pairs of ascents/descents between the lowest safe flight altitude (typically 200 to 500 ft agl) and the flight altitude at which significant changes are observed in measured species concentrations (e.g., $CH_4$, water vapor, relative humidity and potential temperature)- typically 3,000 to 10,000 ft agl. Plots of agl altitude versus these species are used to assess the mixing height of
surface emissions. Both $CH_4$ concentrations along the flight path and the mixing height determined from the airborne vertical profiles are used in transport modeling to determine emissions from the entire study area.

[$CH_4$] emissions are computed from each complete circuit of the study area by the aircraft. This is done by comparing the observed and simulated [$CH_4$] enhancement, the increase in [$CH_4$] downwind of the study area relative to a background value, and adjusting emissions within the study area to minimize the absolute error between the simulated and observed ABL [$CH_4$].
The 10th percentile of [$CH_4$] observations in the circuit determines the background. This mole fraction value is subtracted from the observed [$CH_4$] observations, resulting in an estimate of [$CH_4$] enhancements. These observed enhancements are then compared to simulated [$CH_4$] enhancements by matching observation and model at the nearest gridpoints in space and time. Simulated enhancements are split into two categories: study domain enhancements and enhancements originating from outside the study domain. Enhancements associated with sources outside the study domain are subtracted from the observed [$CH_4$]
enhancements, resulting in a set of observations whose enhancements can be directly attributed to emissions within the study domain. The simulated study domain enhancements are then compared to the observed study domain enhancement, and a scalar multiplier is applied to the simulated enhancements to minimize the absolute error between the two datasets. Because the emissions scale linearly with the simulated enhancements, this scalar multiplier, applied to the preliminary emissions estimate within the study area, provides a solution to the emissions within the study domain (Barkley et al., 2017). The solution
for each circuit is merged into a single daily estimate.

To test the uncertainty of the emission rate solution for each flight day, a 1000-iteration Monte Carlo uncertainty assessment was performed, adjusting various parameters to test how they impacted the solution. Through the iterations we examine the impact of various possible sources of error, including uncertainty in the background, uncertainty in the assumed influence from sources outside the domain, and uncertainty in the atmospheric transport. For uncertainty in the background, we select a random





percentile between 5 and 15 to use as the methane background in a flight lap. For uncertainty in sources outside of the domain that are subtracted from the observations, we multiply the "other" enhancement tracer by a random factor between 0.5 and 1.5 to account for the possibility that regional emissions may be incorrect. For uncertainty in the transport, the time of the observations are adjusted by ± 30 minutes. creating perturbations to the model output timeframe used to compare to the observations. From the 1000 iterations, the 2.5th and 97.5th percentile of solutions are chosen to represent the 95% confidence
interval.

### 2.2.3 Tower-based methane emissions estimates

Atmospheric mole fraction measurements of $CH_4$ and $CO_2$ were collected at five locations in the Permian Basin beginning 1 March 2020, using methods similar to those described in Richardson et al. (2017). A map of the measurement locations, along with oil and gas facilities in the Permian Basin, is shown in Fig. 1.  Note that only four of the five planned measurement sites are used in this analysis and shown on Fig. 1 due to instrument malfunctions at the northernmost site. Of these measurement
locations, three were on towers at measurement heights of 91 – 134 m agl and the westernmost site was at a mountaintop station on a rooftop 4 m agl.  The measurements were made with wavelength-scanned cavity ring down spectroscopic instruments (Picarro, Inc., models G2301, G2401, G2204, and G2132-i).  The air samples were dried using Nafion dryers (PermaPure, Inc.) in reflux mode, with an internal water vapor correction applied for the effects of the remaining water vapor
(< 1 %).  The instruments were calibrated in the laboratory prior to deployment and using quasi-daily field tanks traceable to the WMO X2004A scale (Dlugokencky et al., 2005;NOAA, 2015).  The $CH_4$ measurement uncertainty (including instrument noise, uncertainty due to water vapor calibration and tank assignment uncertainty) for the four tower locations was 0.6 ppb (Carlsbad), 0.6 ppb (Fort Stockton), 3.4 ppb (Hobbs), and 5.4 ppb (Notress), with the differences being attributable to different instrument type and short Nafion dryer in the case of Hobbs, and laser aging (Notrees).

[$CH_4$] emissions in the study domain were calculated for each day of tower observations using a similar technique as used with the aircraft observations. Daily afternoon [$CH_4$] at each tower site averaged from 16Z (11 LST) through 22Z (17 LST) was computed from both the observations and the simulation. A background [$CH_4$] value (both for the observations and the model) is selected based on the lowest measurement from the available tower sites.  This background is subtracted from all tower sites to create an observed $CH_4$ enhancement. Simulated enhancements from sources outside of the domain are subtracted from the
observed enhancements to produce an observed [$CH_4$] enhancement associated with sources inside the study domain. A scalar multiplier is then applied to minimize the absolute error between the observed and modelled enhancements, and a daily emission rate is solved for in the study domain (Fig. 3).

Figure 4 presents the daily difference between the highest and lowest observed $CH_4$ measurement across the tower network. Although though the overall magnitude of the study area plume observed at the tower network can be affected by various
meteorological factors (e.g. wind speed, direction, boundary layer height) large changes in the typical size of the observed plumes can be indicative of sudden shift in behavior of local emissions. From the tower network, we frequently observe large

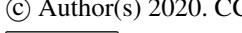



enhancements >200 ppb in March and mid-April, after which point the enhancement rarely increases above 150 ppb for the remainder of the summer months. It should be noted that a slight decrease in the size of the enhancements would be expected during this period due to increased vertical mixing in a seasonally growing boundary layer; however, modelled results from
this timespan exhibit a much smaller magnitude of change. Therefore, the dramatic rechange in behavior $CH_4$ enhancements coincident with the timing of the price crash is likely due to a change in the emissions rather than a change in the meteorology.

Unlike the aircraft mass balance observations, which are collected on days where meteorological conditions are ideal for measuring emissions from the study domain, the tower dataset is continuous and many days may not be suitable for calculating an emission rate from the study domain. The most useful tower observations for solving for emissions within the study domain
are those whose enhancements are influenced primarily by sources within the study domain and contain minimal enhancements from sources outside of the domain. We select for these conditions by retaining days when >50% of the simulated downwind afternoon tower enhancements come from sources within the study domain. This filtering removes 85 of 184 available days, most of which have easterly winds and contain air masses heavily influenced by oil and gas basins in central and eastern Texas. For the remaining 99 days, we remove 5 days whose solutions are more than three median absolute deviations away from the
median solution, presumably caused by issues in the model transport. In total, 94 days are used to calculate emissions and trends in the tower dataset between March 1st, 2020 and August 30th, 2020.

Figure 3 presents a timeseries of $CH_4$ emissions within the 100 x 100 km study area between March 1st, 2020 and June 30th, 2020. This timespan was chosen to focus discussion on the correlations between emissions and activity in the basin during a period of volatile oil prices; accurate activity data is only available through the month of June at the time of manuscript
submission (Appendix E). Figure 4 extends the observation of the tower and aerial measurements from Fig. 3 through August 30th 2020 while also presenting monthly mean estimates and 95% CI of the combined results from aircraft and tower-based estimates (Table 1). The 95% CI ranges are derived from twice the standard error of all accepted estimates in each month. Extending the dataset reveals that the mean emissions decreased briefly in July before increasing in August to a similar level as observed in March 2020, albeit with larger uncertainty. Without production or other activity data to relate the observations,
we do not have a mechanistic understanding of the reason for the decreased emissions in July, although we note the significant overlap in confidence intervals does not allow for statistically independent estimates between the months of June, July, and August.



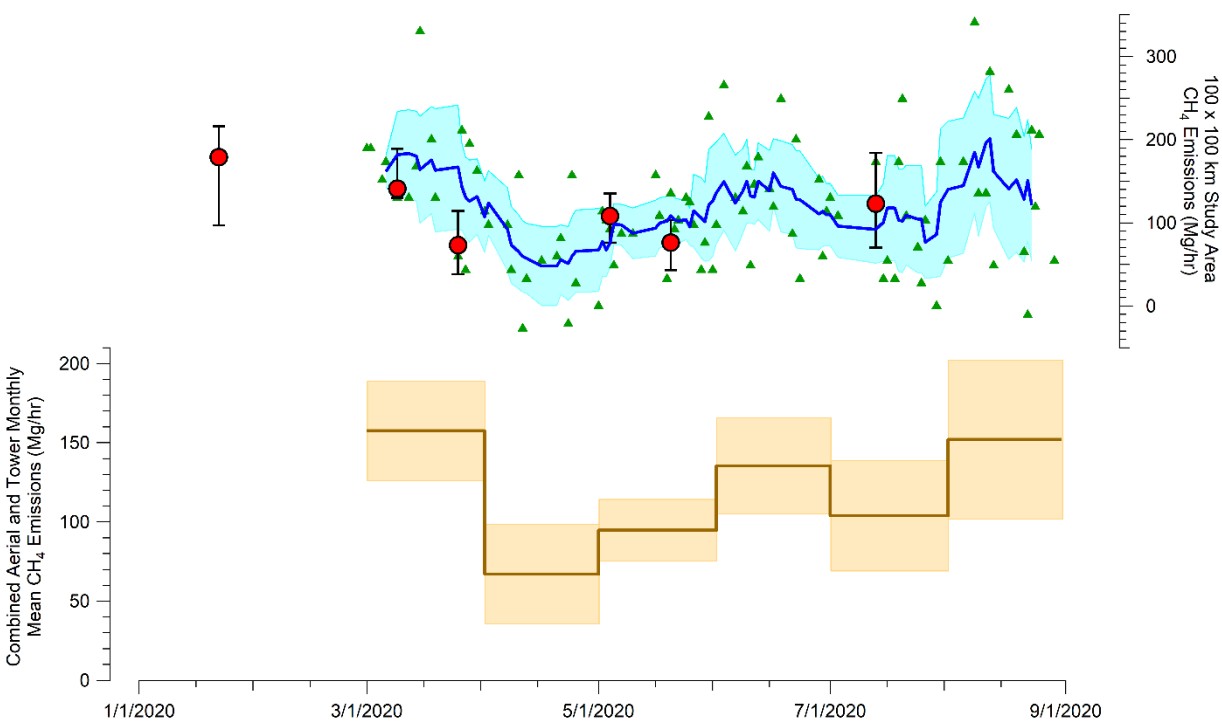

**Figure 3.** Tower and aerial emission estimates from the 100 x 100 km study area extended for the full available record through
August 2020. Individual daily accepted estimates from the tower observations are shown in green triangles while red points
represent the aerial estimate and 95% CI and Blue line and shading represent the 7-datapoint moving average of the tower
estimates and 95% CI expressed as twice the 7-datapoint moving standard error. Brown line and shading on the lower plot
represents the monthly mean estimate and 95% CI from the combination of aerial and tower-based methods.


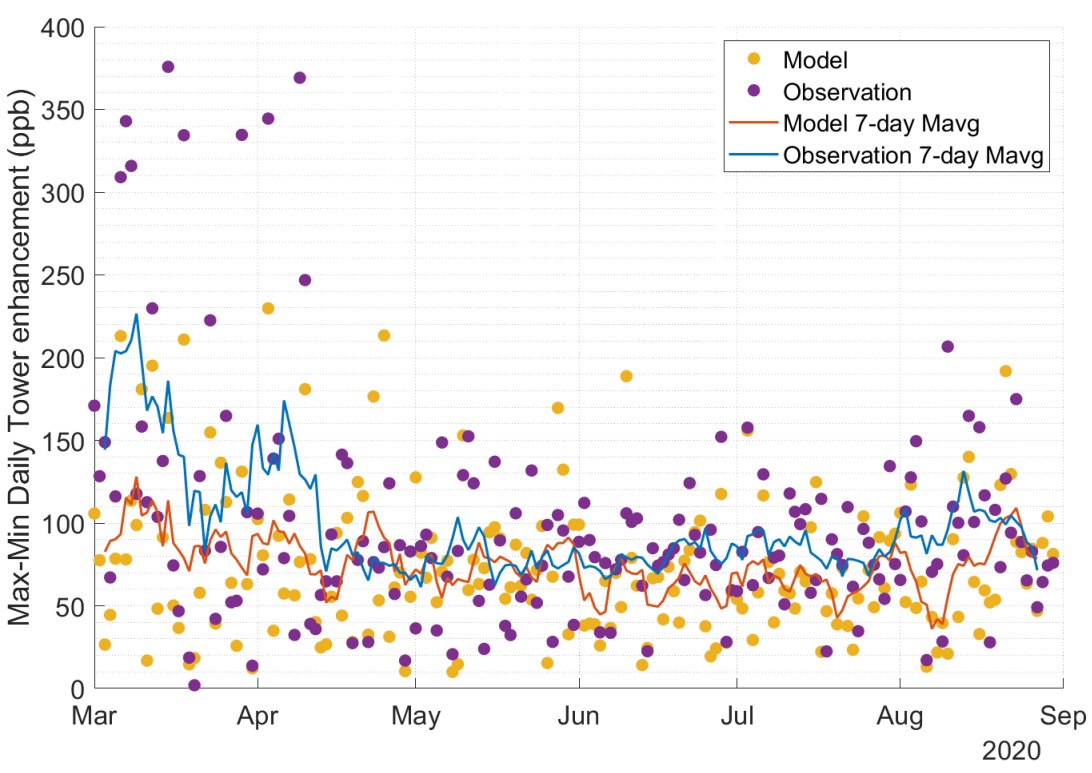

**Figure 4**. Comparison between modeled and observed differences in the maximum and minimum daily $CH_4$ enhancement across the tower network. Also shown are the 7-day moving averages of each trend.



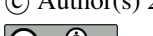



**Table 1.** Numerical estimates of CH$_4$ flux from the 100 x 100 km study area derived from the combination of tower and aerial measurements across several temporal ranges.

| Time Range | Mean Emissions (Mg/hr) | Standard Deviation (Mg/Hr) | Number of accepted daily tower and aerial measurements | Standard Error (Mg/hr) | 95% CI Emission estimate |
|---|---|---|---|---|---|
| March 2020 | 157 | 66 | 17 | 16 | 126 - 189 |
| April 2020 | 67 | 58 | 13 | 16 | 35 - 99 |
| May 2020 | 95 | 47 | 22 | 10 | 75 - 115 |
| June 2020 | 135 | 64 | 17 | 16 | 105 - 166 |
| July 2020 | 104 | 69 | 15 | 18 | 69 - 139 |
| August 2020 | 152 | 100 | 15 | 26 | 101 - 202 |
| 'Pre-Crash Period'  Jan 22 - Mar 19 2020 | 176 | 59 | 12 | 17 | 142 - 209 |
| 'Emissions Minima' Apr 7 - May 1 2020 | 55 | 61 | 12 | 17 | 21 - 89 |




### 2.2.4 TROPOMI-derived column-averaged methane mixing ratios

We use column-averaged dry air methane mixing ratios ($XCH_4$) from the TROPOMI instrument from January to June 2020. TROPOMI was launched in October 2017 onboard the polar sun-synchronous Sentinel-5 Precursor satellite with an ~13:30

local overpass time. It provides daily global coverage with 7 km x 7 km pixel resolution at nadir (Hu et al., 2018); the pixel resolution has changed to ~7 km x 5.5 km at nadir since August 2019. The $XCH_4$ retrieval uses sunlight backscattered by the Earth's surface and atmosphere in the shortwave infrared (SWIR) spectral range and has near-unit sensitivity down to the surface (Hasekamp et al., 2019). Here we consider only higher-quality $XCH_4$ measurements (quality assurance value > 0.5).

Figures 5a and 5b show mean methane column enhancements over the Permian basin, observed by TROPOMI in (a) January-

February 2020 and (b) April-May 2020. We calculate the daily methane enhancements over the Permian basin from topography-corrected $XCH_4$, relative to a regional background column defined by the $10^{th}$ percentile of $XCH_4$ across the full Permian domain (29-34°N, 100-106°W). The topography correction is based on a linear regression of $XCH_4$ against surface altitude (similar to the methodology presented in (Kort et al., 2014;Zhang et al., 2020), performed across the continental United States (25-48°N, 66-125°W). Enhancements over the Permian basin appear to be lower in April-May compared to January-

February, as indicated by an ~18% reduction in the regional mean between those two periods. This reduction may be due in part to lower spatial coverage after February 2020, likely caused by the introduction in March of a different cloud mask product in the TROPOMI retrieval algorithm (Siddans, 2020). Considering TROPOMI retrievals with quality assurance values of 0.5 or greater, we obtain roughly 6,000-32,000 enhancement measurements per month from January to June 2020 over the full Permian Basin (Fig. 5c). The limited number of satellite observations over our 100 km x 100 km study area for tower and

aircraft measurements (Fig. 3) precludes direct comparison with the suborbital measurements, and therefore we provide here an analysis of TROPOMI methane enhancement over the broader Permian Basin. Coverage is particularly sparse in March and June, so we neglect those two months in the TROPOMI analysis presented here.

Figure 5d shows frequency distributions of methane column enhancements observed by TROPOMI in January, February, April, and May 2020. For these monthly curves we restrict our attention to a smaller Permian domain that closely bounds the

methane hotspots seen over the Delaware and Midland sub-basins (dashed lines in Figs. 5a,b; 31-34°N, 101.4-105.6°W). Roughly 5,000-14,000 TROPOMI observations are available per month across this domain, neglecting March and June (Fig. 5c). To mitigate the impact of reduced spatial coverage on our change analysis after February, we manually discard observations from days with little to no coverage of the Delaware and/or Midland sub-basins. Data from 20-40% of observation days in January, February, April, and May (depending on the month) are discarded in this way, but the total number of

observations is reduced by only 5%. Permian basin methane enhancements as observed by TROPOMI appear to decrease in early 2020, reaching a minimum in April before beginning to rise again in May.

Repeating our analysis with the background defined at the $25^{th}$ percentile level (rather than the $10^{th}$), we find that these trends are insensitive to the percentile value used. Furthermore, the trends are not explained by seasonal changes in wind speed across



the Permian. Higher winds could lead to lower enhancements, but data from the NASA GEOS-FP meteorological reanalysis
product indicate that the daily wind speed averaged over the full Permian basin domain, in the lowest 3 km of the atmosphere, during the six hours closest to TROPOMI observation time (15:00-21:00 UTC) decreased from a mean of 7.02 m/s in January-February to 5.48 m/s in April-May.

The trends we identify in TROPOMI methane enhancement analysis across the Permian Basin are broadly consistent with our findings from tower and aircraft observations of reduced emissions particularly during April in our campaign domain of the
Delaware sub-basin, but large uncertainties remain due to the different spatial domains and the reduced satellite coverage after February 2020. More data and/or more advanced analysis using inverse modelling techniques may be needed to reliably characterize Permian basin methane emission trends using TROPOMI satellite observations.


**Figure 5.** TROPOMI observations of topography-corrected methane column enhancements over the Permian basin, from January to June 2020. a-b: Mean methane column enhancements (ppb) over the Permian basin for the January-February and April-May 2020 time periods, gridded to 0.1° x 0.1° resolution. The thin solid lines indicate state and national borders; the

thick solid lines describe the 100 km x 100 km tower and aircraft study region; and the dotted lines trace a smaller Permian domain that closely bounds the methane hotspots seen over the Delaware and Midland sub-basins. c: Number of TROPOMI column retrievals over the full Permian basin domain (29-34°N, 100-106°W) and over the smaller Permian basin domain (31-34°N, 101.4-105.6°W; dashed lines in panels a, b), by month in 2020. d: Frequency distribution plots of methane column enhancements over the smaller Permian domain, by month, after removal of days without coverage of the Delaware and/or

Midland sub-basins (see text). The gray vertical line indicates the distribution maximum for January.





## 3 Results

Both aircraft and tower-based methane flux data show consistent trends of declining then rebounding methane emissions in our Permian Basin study area during March – June 2020 (Fig. 6). Between January 22 and March 19, 2020, Emissions were

176 Mg $CH_4$ $hr^{-1}$ (95% confidence interval range: 142 – 209 Mg $CH_4$ $hr^{-1}$). Following the rapid decrease in oil price, emissions between April 7 and May 1, 2020 reached a minimum of 55 Mg $CH_4$ $hr^{-1}$ (95% CI range: 21 – 89 Mg $CH_4$ $hr^{-1}$).  After the oil price mostly recovered, emissions for the month of June had increased to 135 Mg $CH_4$ $hr^{-1}$ (95% CI range 105 – 166 Mg $CH_4$ $hr^{-1}$) and by August had recovered largely to pre-crash levels at, albeit with higher uncertainty (152 Mg $CH_4$ $hr^{-1}$; 95% CI range 101 – 202 Mg $CH_4$ $hr^{-1}$, Fig. 3). Combining the monthly tower estimates with reported gas production (Enverus, 2020), we

calculate a March 2020 loss rate of 3.4% of total gas production (95% CI range: 2.7 – 4.1%), slightly lower but within the uncertainty of previously reported basin wide estimates from 2018 – 2019 (3.7 ± 0.7 (1σ) %) (Zhang et al., 2020). The minimum loss rate calculated for April 2020 was 1.5% of gas production (95% CI range: 0.8 – 2.2%); the loss rate of later months is uncertain due to incomplete gas production data (Appendix E). In the full Permian Basin, orbital observations of $XCH_4$ indicate lower methane column enhancements in April – May versus January – February 2020, consistent with the aircraft and tower-

based flux data (Fig. 5).

Well pad development in the study area proceeded at an average rate of 71 new sites per month between August 2019 and March 2020, then dropped to a monthly average of 24 sites between April and July 2020 (Appendix C, Fig. 7). The number of well completions per month declined from 134 to 53 between January and April 2020 (Enverus, 2020); completion counts are higher than well pad development rates due to multiple wells being located on a single pad. After rising steadily throughout

2019, oil and gas production peaked in March 2020 and then declined 9 and 8%, respectively, in April. Based on adjusted, incomplete production data for May and June, gas production stayed relatively steady after April while oil production dropped an additional 3% (Appendix E). The relative decline in O&G production between March and April 2020 was much greater among wells in the first two months of production, decreasing 50 and 45%, for oil and gas, respectively (Appendix E).

The three flare surveys between February and June 2020 consistently found that 11% of flares had combustion issues, with

5% unlit and emitting hydrocarbons. Even when using conservative assumptions of greater combustion efficiency, we estimate a basin-wide flare combustion efficiency of 93%, with the remaining gas (assuming 80% methane content) being emitted to the atmosphere (Appendix B). Satellite observations of radiant heat indicate that flared gas volumes were cut in half from 7.6 to 3.2 Bcf between January and April 2020 (Fig. 8).





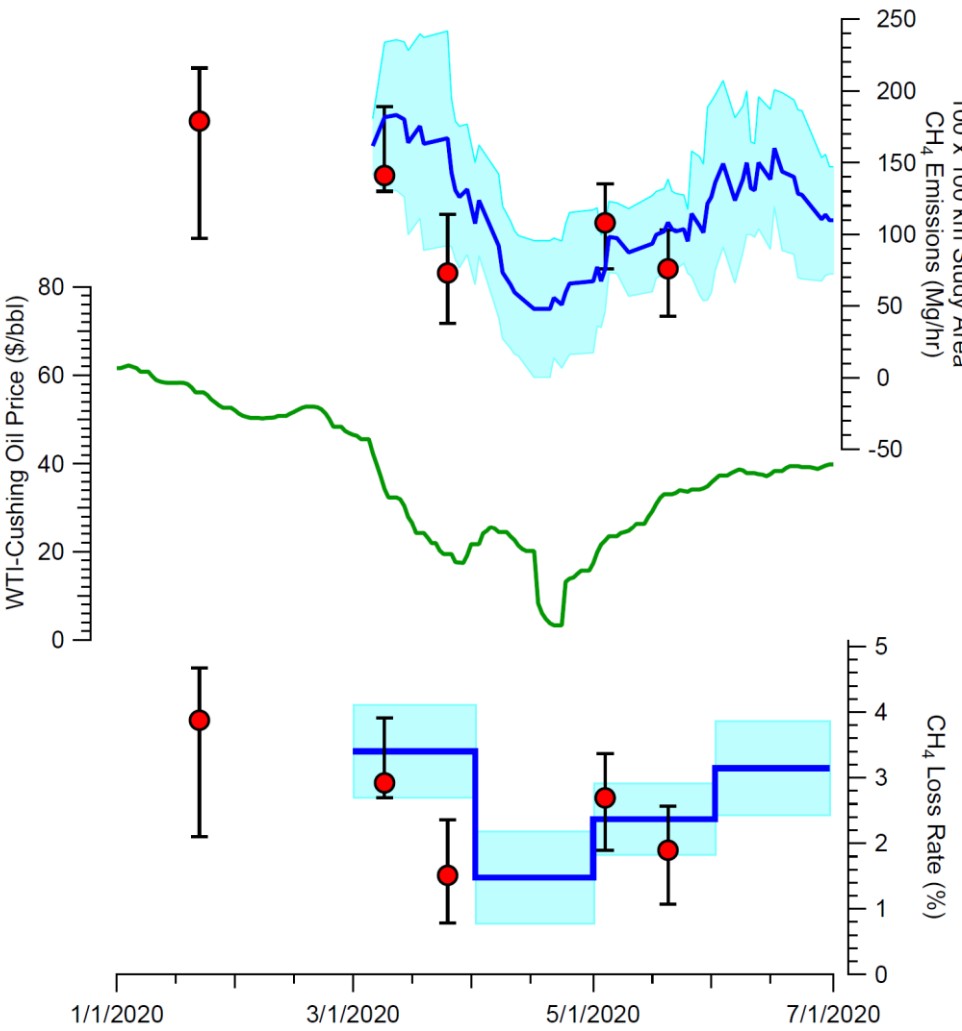

**Figure 6.** Temporal variation in methane emissions and crude oil price. Top: Aerial (red circles with 95% CI error range) and tower based 7-point moving average and 95% CI (blue line and shading) atmospheric estimates of 100 km x 100 km study area $CH_4$ emissions. Middle: 7-day moving average of WTI-Cushing daily oil price. Bottom: Resulting $CH_4$ loss rate from aerial and monthly mean tower-based measurements utilizing published monthly gas production within the study area (Enverus, 2020).


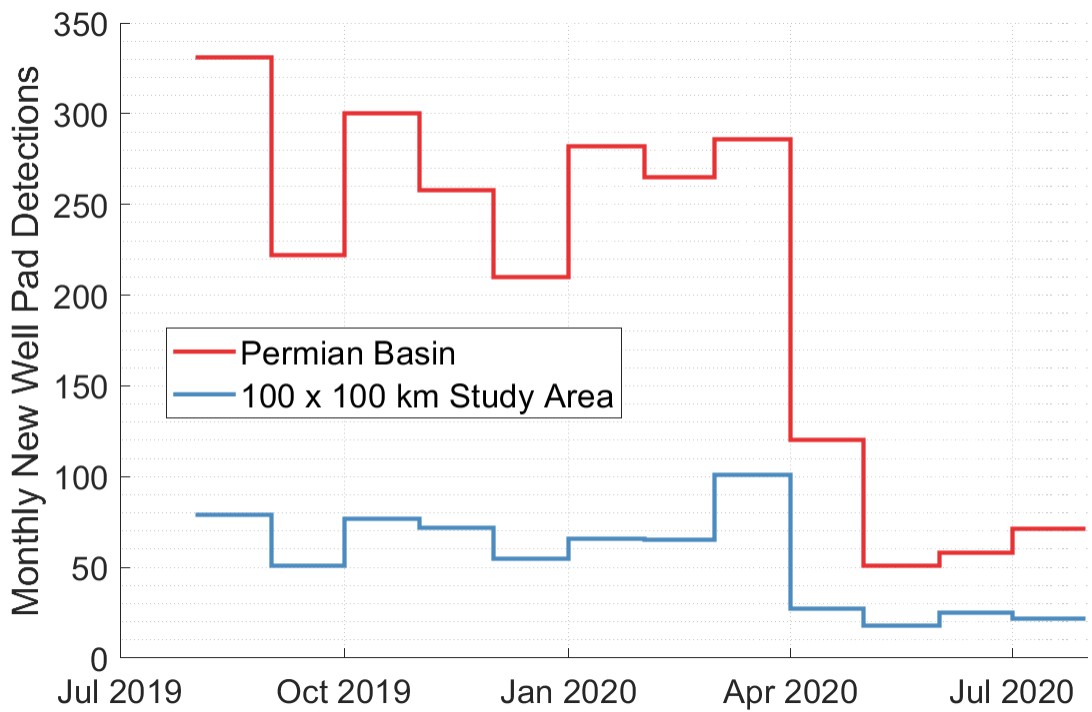

**Figure 7.** Number of new well pads constructed per month between August 1, 2019 and August 1, 2020.





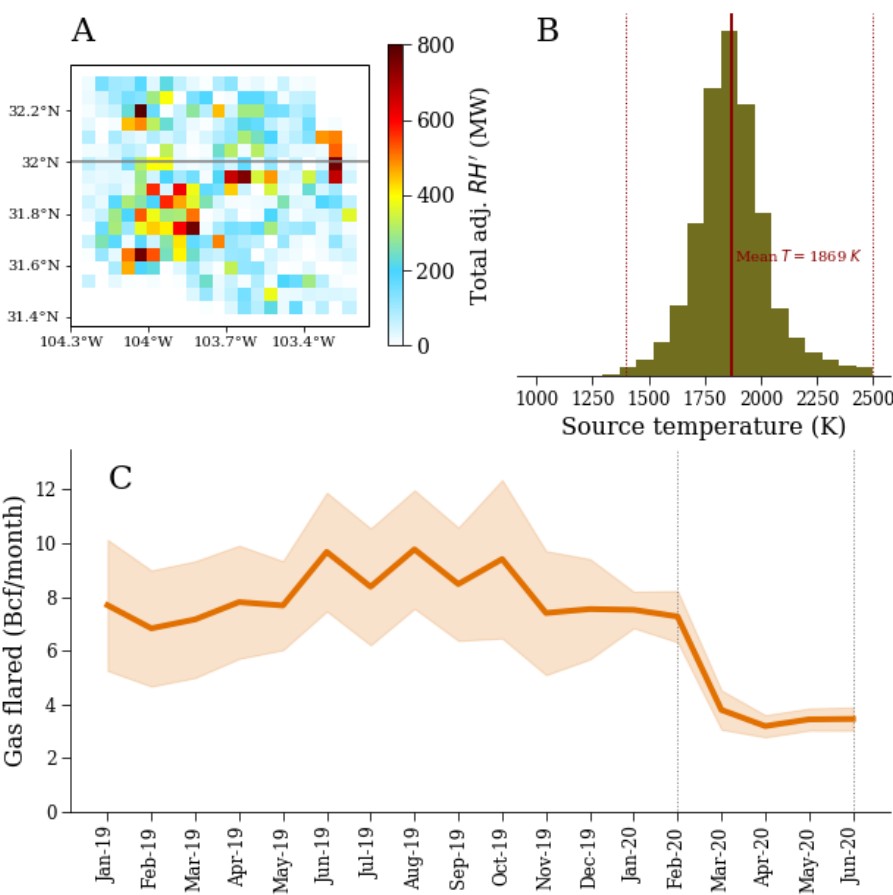


**Figure 8.** VIIRS-derived gas flaring in the study region. (A) Spatial distribution of the cumulative adjusted radiant heat over the period between January 2019 and June 2020 aggregated over a 0.05 x 0.05° grid resolution. (B) Histogram of VIIRS-derived source temperatures. Dotted lines show the temperature regime characteristic of gas flaring sources (1400—2500 K). (C) Monthly trend in VIIRS-derived gas flared volumes. The mean estimate in shown in solid line and the 95% CI on the mean is shown in the shaded area.





## 4 Conclusions

The pandemic-related oil price crash provided an unexpected opportunity to assess temporal variability in methane emissions during a period of volatile oil prices and associated operational changes. In support of our hypothesis that methane emissions would decline with oil price, we observed a three-fold reduction in Permian Basin study area methane emissions that was strongly correlated to the average daily oil price. The relative decline in O&G production during this period was less than 10%; accordingly, loss rate temporarily decreased from 3.4% to 1.5% of gas production between January 22 – March 19 and April 7 – May 1, 2020 (Appendix E). It is important to note that even the minimum observed loss rate of 1.5% is several times higher than the performance targets committed to by major O&G companies accounting for about one-third of global oil production, including some with operations in the Permian Basin (OGCI, 2020). We hypothesize that total methane emissions are positively correlated with oil price due to three interrelated factors associated with well development: 1) well completion rates, 2) associated gas flaring volumes, and 3) indirect impacts of new gas production on the gathering and processing (G&P) system.

Lower oil prices directly led to reduced emissions by decreasing well development activities, as we observed for rig count, new site construction, and well completions following the price crash. Well development activities are an intermittent source of methane emissions, particularly completion flowback, the typically multi-day period following hydraulic fracturing when fluids, excess proppant, and entrained gas are expelled from the wellbore (Allen et al., 2013). We estimate that the ~80 fewer well completions in April versus January 2020 caused average potential flowback emissions in our study area to decline from 9 to 2 Mg $CH_4$ $hr^{-1}$ (Appendix D). At the time of the study, U.S. federal regulations mandated the use of reduced emission completions to control emissions in most situations; however, operator reported data suggest actual emissions (1 – 2.5 Mg $CH_4$ $hr^{-1}$) are of similar magnitude to our estimate of potential emissions. ((USEPA, 2019, 2020b); Appendix D).

The observed two-fold reduction in flared gas volumes between January and April 2020 was likely the result of the large drop in gas production from new wells. Unconventional wells tend to have high initial gas production followed by steep declines. With lower rates of well development and new gas production in the area, competition for limited gas pipeline capacity likely was abated, leading to less flaring of stranded associated gas. Assuming a combustion efficiency of 93%, we estimate flare-related methane emissions in our study area were approximately 8 and 3 Mg $CH_4$ $hr^{-1}$ in January and April 2020, respectively (Appendix A). Our combustion efficiency assumption, which is based on repeat observations of over 300 flares, is conservatively high and therefore our emission estimate represents a lower bound. However, even with worst-case assumptions of flare combustion efficiency it is unlikely that January and April flare-related emissions would have exceeded 20 and 7 Mg $CH_4$ $hr^{-1}$, respectively (Appendix B).

Our estimates of well completion and flare-related methane emissions account for less than 20% of the observed total reduction between pre-crash and minimum price conditions; therefore, we theorize that the primary driver of emission reductions is indirect improvements to G&P system performance resulting from reduced inputs of gas from new wells. This result suggests



that the high methane emission rate observed in the Permian Basin in recent years is in large part due to insufficient capacity of G&P infrastructure for handling and delivering rapidly growing rates of natural gas production (Zhang et al., 2020). The drastic decline in flared gas volumes during the oil price crash suggests that the reduction in new gas production relieved G&P capacity issues. A similar pattern was observed in the Bakken formation during the oil price decline of 2015-2016: price drops caused only a small decrease in total production but a large decrease in drilling and flaring rates (Appendix F). Our study provides the first direct evidence of reduced methane emissions resulting from an apparent abatement of infrastructure capacity

limitations.

The high methane emission rate observed in the Permian Basin during periods of higher commodity prices is likely a consequence of associated gas production increasing at a faster rate than midstream infrastructure capacity, which leads to extensive flaring and anomalous conditions related to excess gas throughput (e.g. pressure relief venting). Our observations of emissions declining concurrently with new well development suggest that methane emissions could be mitigated in the Permian

Basin and similar oil-producing fields by better aligning development rates of wells and midstream infrastructure. For example, regulations could prohibit the drilling of wells in areas without sufficient capacity to transport newly produced gas to market. Our findings suggest that policies which tie the maximum rate of well development to infrastructure capacity, in addition to other approaches such as requiring high frequency or continuous monitoring to detect large emission sources (Alvarez et al., 2018), can facilitate lower methane emissions that reduce the climatic impact of oil and gas production.

**Appendix A. VIIRS-derived flared natural gas volumes**
We assess the monthly trends in the volumes of natural gas flared in the study region using nighttime fire and flare data observed by the Visible Infrared Imaging Radiometer Suite (VIIRS) instrument onboard the Suomi National Polar-Orbiting Partnership satellite. Specifically, we use the VIIRS NightFire V3.0 data product to support our analysis (Elvidge et al., 2013) For the study region and for the period between January 2019 and June 2020, we retrieved 49,885 individual VIIRS detections

for which it was possible to estimate flaring source temperatures based on Planck curve fitting of the source radiances Elvidge et al. (2013). During this period, the mean VIIRS-derived source temperature was 1869 K. The histogram of source temperatures is shown in Fig. 8b, indicating a strong gas flaring signal in the characteristic temperature regime of between 1400 and 2500 K. Elvidge et al. (2015) developed a correlation between the VIIRS-derived radiant heat and reported gas flared volumes and derived the relationship:


$$V_a = 0.0274\, RH'\,(R^2 = 0.86)$$

where $V_a$ is the annual volume of gas flared (in billion cubic meters) and $RH'$ is the modified radiant heat for each individual flare, adjusted to account for the observed non-linear relationship between flared gas volume and radiant heat and was

computed as: $RH' = \sigma T^4 S^{0.7}$ , where $\sigma$ is the Stefan-Boltzmann constant (5.67 x $10^{-8}$ W m$^{-2}$ K$^{-1}$), $T$ and $S$ are the source temperature and area, respectively, and the exponent (0.7) was empirically developed by Elvidge et al. (2015) to address non-



linearity. Figure 8a shows the spatial distribution of the cumulative $RH'$ in the study region over the period between January 2019 and June 2020, as aggregated over a $0.05° \times 0.05°$ grid resolution. To estimate monthly gas flared volumes ($V_m$ in billion cubic feet) for the study area, we modify equation the equation above, assuming the relationship holds over monthly intervals:


$$V_m = 0.0274\, RH' \times \frac{1}{12} \times 35.315\, [\frac{Bcf}{Bcm}]$$

We use the equation above to compute the mean monthly gas flared volumes (and 95% CI on the mean) in the study area based on the daily $RH'$ aggregated from individual detected flares. The trend in the monthly gas flared volumes is shown in Fig. 8c.

The average flaring rate in 2019 was $8.2 \pm 2.2$ Bcf/month. From February 2020, a sharp decline in the mean gas flaring rate was observed, with the lowest estimated flaring rate of $3.2 \pm 0.4$ Bcf in April. Following a similar procedure for the entire Permian region, the estimated mean monthly flaring rate declined from a mean of $23 \pm 5$ Bcf/month in 2019 to $8.1 \pm 1.7$ Bcf in May 2020. Thus, the lowest estimated monthly gas flared volumes in 2020 were a factor of 2.6 and 2.8 times lower than the monthly mean observed in 2019 for the 100 km x 100 km study region and full Permian Basin, respectively.


### Appendix B. Aerial flare performance survey

We compiled a list of potential locations of recently active flares in the Permian Basin (Delaware and Midland sub-Basins) based on a geospatial analysis of the SkyTruth Global Flaring Dataset, which is derived from heat sources detected by the Visible Infrared Imaging Radiometer Suite (VIIRS) instrument on the NOAA Suomi NPP satellite; SkyTruth has applied

several filters to the VIIRS data including removing heat sources <1,500 ℃ and with <3 detections per month (Skytruth, 2020). To account for spatial uncertainty of SkyTruth flare locations, we spatially joined their individual flare detections between October 1, 2019 and January 31, 2020 using a 100-meter buffer distance; the centroid latitude/longitude of the 1,014 joined detections were defined as likely locations of recently active O&G flares. Leak Surveys, Inc. (LSI), a leak detection company specializing in aerial optical gas imaging, was provided a list of 573 potential active flare locations from the original set of

1,014. The site selection methodology balanced representativeness and survey efficiency by defining one contiguous, high flare density area in each sub-basin that could be surveyed over the course of approximately five days. For the Delaware sub-Basin, we selected 323 locations located within our main study area (NW and SE corners are 32.325° N, 103.822° W and 31.417° N, 103.202° W, respectively). For the Midland sub-Basin, we selected 250 locations from the two counties (Midland and Martin) with the highest flare counts from the analysis of VIIRS data. LSI surveyed these locations with a custom infrared

camera (IR) deployed in a R44 helicopter. Potential flare locations were identified with spatial coordinates and a unique flare ID.

LSI performed three surveys of the potential flare locations during the weeks of February 17, March 23, and June 22, 2020 (EDF, 2020). At each potential flare location, LSI determined if one or more flares was present at the spatial coordinates, and





if so, observed the flare(s) for operational status. For flares with apparent combustion issues, LSI recorded 30 – 60 seconds of infrared and visual video footage of the flare plume to provide visual evidence of flare status. For each flare, LSI assigned a qualitative assessment of the apparent flare status at the time of survey from four categories: inactive and unlit with no emissions (inactive); active, lit, and operating properly (operational); active and lit but with operational issues such as incomplete combustion or excessive smoke (malfunction); or active, unlit, and venting methane (unlit). For survey 1, LSI

observed 337 flares from the random selection of potential locations. For surveys 2 and 3, a random subset of the 337 flares was selected for re-survey, prioritizing locations that had previously observed issues. We observed similar flare performance in each of the three surveys: 11% of active flares had observed malfunctions, including 5% that were unlit and venting (Table B1).

To estimate methane emissions from flaring, we used our qualitative flare performance data and conservatively high assumptions about the combustion efficiency of operational, malfunctioning, and unlit flares to estimate overall combustion efficiency, and then applied combustion efficiency to estimated flared volumes in 2019 based on an analysis of VIIRS data (Appendix B). We assume that operational flares perform at the EPA default combustion efficiency of 98% (Regulations, 2016). The 5% of flares that were unlit and venting were assumed to have a combustion efficiency of 0%. The 6% of flares

that were lit with apparent combustion issues were assumed to have 90% combustion efficiency. If we assume flared gas volumes are proportional to the observed fraction of flares by performance, then the overall combustion efficiency of active flares in the Permian Basin is 93%, which means 7% of flared methane is emitted. Applying 93% combustion efficiency to the 280 Bcf of gas flared in the Permian in 2019 (assuming 80% $CH_4$ content) results in annual methane emissions of approximately 300,000 Mg $CH_4$ from flaring in the Permian; unlit flares account for about 65% of these emissions, while

operational and poorly combusting flares account for about 15 and 10%, respectively. As a sensitivity analysis, we use alternative combustion efficiency assumptions of 90%, 50%, and 0% for operational, malfunctioning, and unlit flares, respectively; this leads to an overall combustion efficiency of 83% and 2.3x more flare-related methane emissions that our conservatively low assumptions.

EPA publishes two separate estimates of Permian flaring methane emissions, which incorporates the 98% combustion efficiency but different gas flared data. The 2020 Greenhouse Gas Inventory (USEPA, 2020a) reports 2018 Permian Basin methane emissions of 12,100 Mg $CH_4$ from associated gas flaring, plus 8,500 and 4,600 Mg $CH_4$ from associated gas venting and miscellaneous production flaring, respectively. The Greenhouse Gas Reporting Program (USEPA, 2020b) reports 18,800 Mg CH4 from Permian Basin onshore production facilities.






**Table B1.** The operational performance of Permian Basin flares as observed during three helicopter-based infrared optical gas imaging surveys**.**

| Surveyed Flares | Survey 1 | Survey 2 | Survey 3 | Average |
|---|---|---|---|---|
| Operational | 276 | 147 | 237 | |
| Inactive | 25 | 0 | 62 | |
| Combustion Issue | 23 | 9 | 18 | |
| Unlit and Venting | 13 | 10 | 12 | |
| Total | 337 | 166 | 329 | |
| | | | | |
| Malfunctioning (% of active) | 11.5% | 11.4% | 11.2% | 11.4% |
| Unlit and Venting (% of active) | 4.2% | 6.0% | 4.5% | 4.9% |

**Appendix C. Satellite imagery and machine learning based estimates of well pad development**

We mapped new well pad construction in the Permian Basin using a two-step machine learning and remote sensing approach. First, well pad candidates were identified in satellite imagery with a convolutional neural network (CNN) model in individual scenes. The model predictions were then compared between the beginning and end of each month to identify the locations of newly constructed well pads. Second, by differencing before/after model outputs, persistent false-positives in the model were

removed. The resulting model was deployed on imagery over the Permian Basin on a monthly cadence between August 1, 2019 and July 1, 2020.

We assessed the monthly trends in new well pad construction in the Permian Basin using a combination of satellite imagery from the European Space Agency Sentinel-2 satellite (ESA, 2020) and the National Aeronautics and Space Administration

(NASA) Landsat-8 satellite (USGS, 2020). Imagery from Sentinel-2 has a pixel resolution of 10m, sufficient to clearly identify well pads, and is collected approximately once every 5 days for any location, providing an average of 6 collects per month. While this is generally sufficient for monthly monitoring, some areas experience high cloud cover in all the scenes, causing well pads to be missed. Imagery from Landsat-8 was used to fill in for such cloudy scenes. Despite the slower 16-day revisit rate and coarser (30m) pixel resolution of Landsat-8, well pads are still easily detectable. The combined use of these two

satellites provided at least one cloud-free scene for all of the Permian Basin for each month within the time period we monitored. We use six spectral bands from both Sentinel-2 and Landsat-8: "red", "green", "blue", "NIR", "SWIR1", and "SWIR2".



New well pad construction was detected in a two-step approach. Well pad candidates were first identified with a convolutional

neural network (CNN) model in individual scenes. The model predictions were compared between the beginning and end of each month, and new well pads were identified. Well pads were detected using a semantic segmentation approach. We used a UNet architecture with a six-band input layer with shape *(height, width, 12)* and output predicting the presence or absence of well pads in each pixel. Landsat-8 imagery was resampled to 10m to match the resolution of Sentinel-2 imagery.

The model was trained on a ground-truth dataset taken from well pads detected with a separate machine learning model run on high resolution (1.5m) imagery. We generated ~7000 training tiles, each of size 512 x 512 pixels and containing 0 to 400 well pads each. The dataset was split into sets with 70 % for training, 10% for validation, and 20% for testing. Examples of image-target pairs are shown in Fig. C1.

New well pads were detected by comparing model output heatmaps between the beginning and end of sequential monthly time periods (Fig. C2). Intuitively, pixel values in satellite imagery change frequently in irrelevant ways, so it is more effective to identify change in the model output. The heatmap from the earlier time was subtracted from the later time. A threshold operator followed by a morphological opening operation were applied to these difference maps. New well pad detections were identified in the resulting binary map as shown in Fig. C3.


To further remove false positives, we require that new well pad candidates should not have existed in multiple months leading up to the construction date, and should continue to exist for several months after. We thus used the three months before and the two months after to remove candidates that fail this condition. While the 10m resolution of the imagery makes it difficult to confirm with certainty that candidates contain oil and gas infrastructure, we suspect that the Permian Basin region is unlikely

to experience a high volume of unrelated ground clearing for development. We confirm this with manual inspection, see details below.

The CNN and change detection pipeline was run over the Permian Basin on monthly imagery composites between August 1, 2019 to July 1, 2020. The deployment was done using the Descartes Labs platform. Tiled imagery was drawn on-the-fly, model

inference was performed in a cloud-native kubernetes infrastructure, and results were stored in the commercial cloud. Finally, the authors hand-verified the candidates for each month.

The change detection analysis has a precision of ~100%, since the final results have been hand-verified. It is infeasible to measure the model accuracy or recall directly, as these would require identifying a substantial number of newly constructed

well pads as well as false negatives (newly constructed well pads that were missed by the model), which would require extensive hand-labeling; additionally, the model performance may vary across geographies, making a single metric less useful. Instead, we estimated the recall using a dataset of well pads identified with a separate machine learning model in high-




resolution imagery; we measured the fraction of these well pads that are detected as well pads by the UNet in single mosaics. Any well pads missed in this step will not be identified as new well pads. We measured this recall on four separate monthly

mosaics, and found a recall of 90.0%, with a statistical uncertainty of less than a percent. Finally, the number of newly constructed well pads per month are shown in Fig. 7 with examples presented in Figs. C4 and C5.

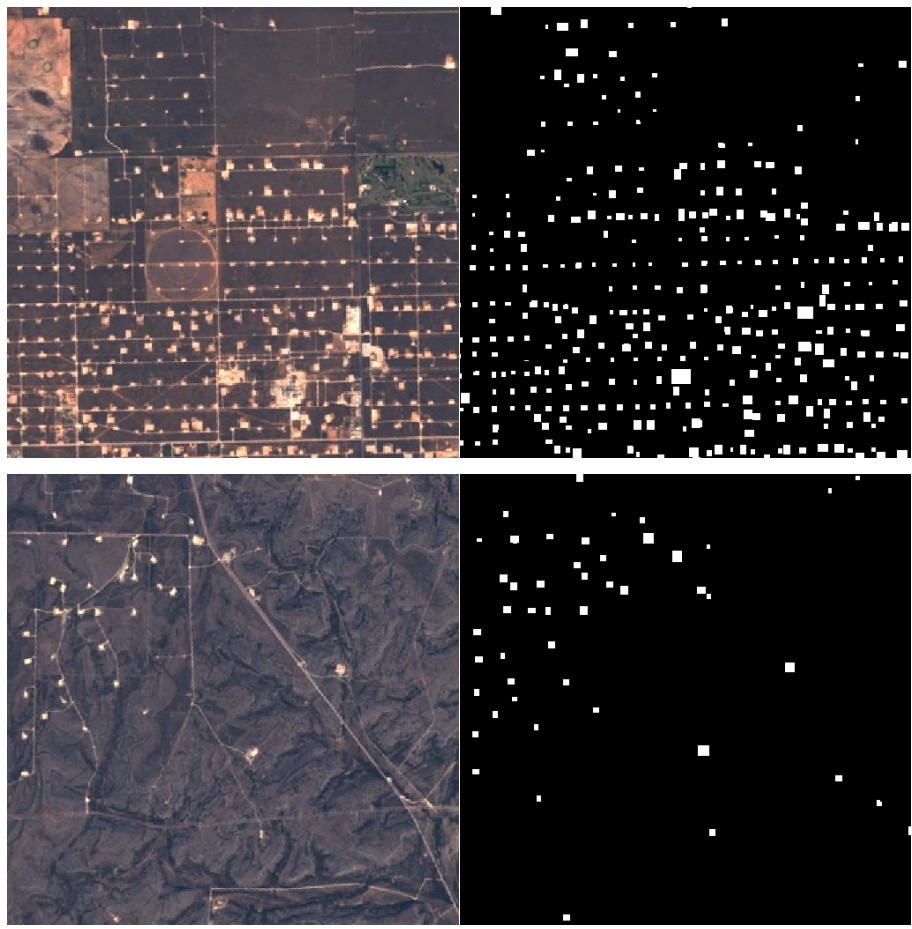


**Figure C1.** Examples of image-target pairs: (left) Sentinel-2 RGB imagery (ESA, 2020); (right) Ground truth






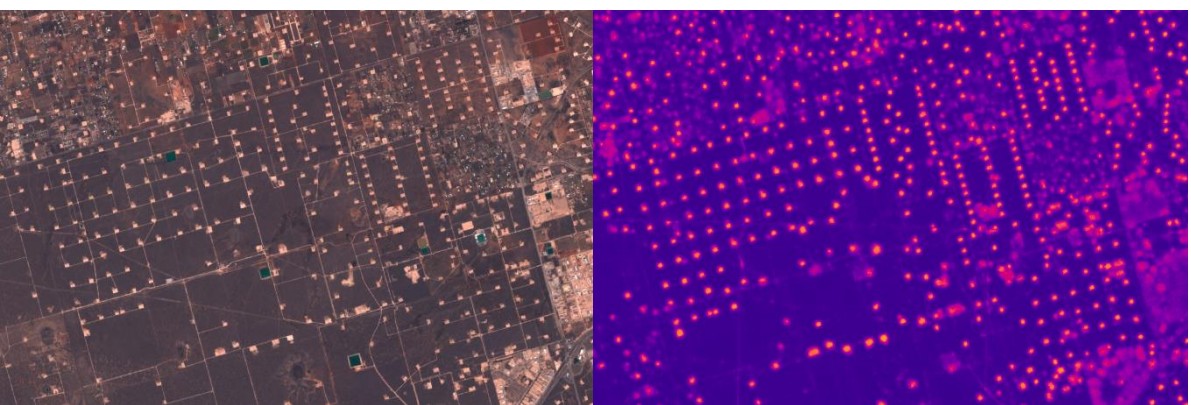

**Figure C2.** CNN model example, showing Sentinel-2 imagery (left; ESA, 2020) and model output heatmap over the same area (right).



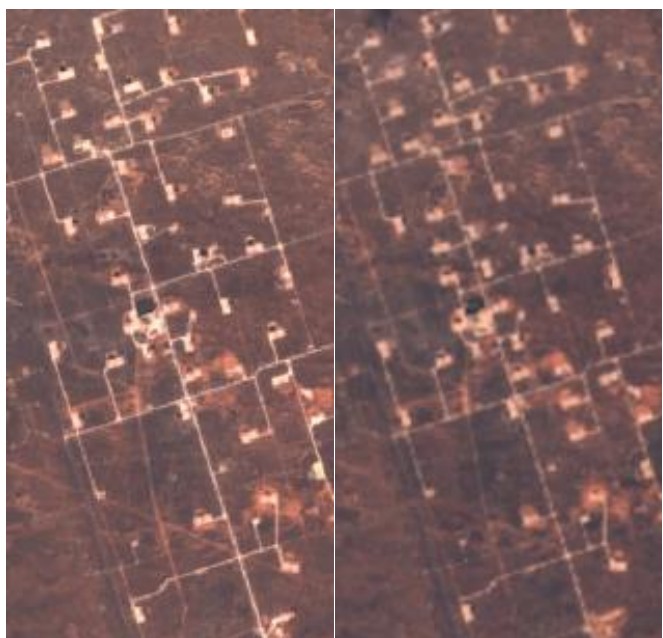


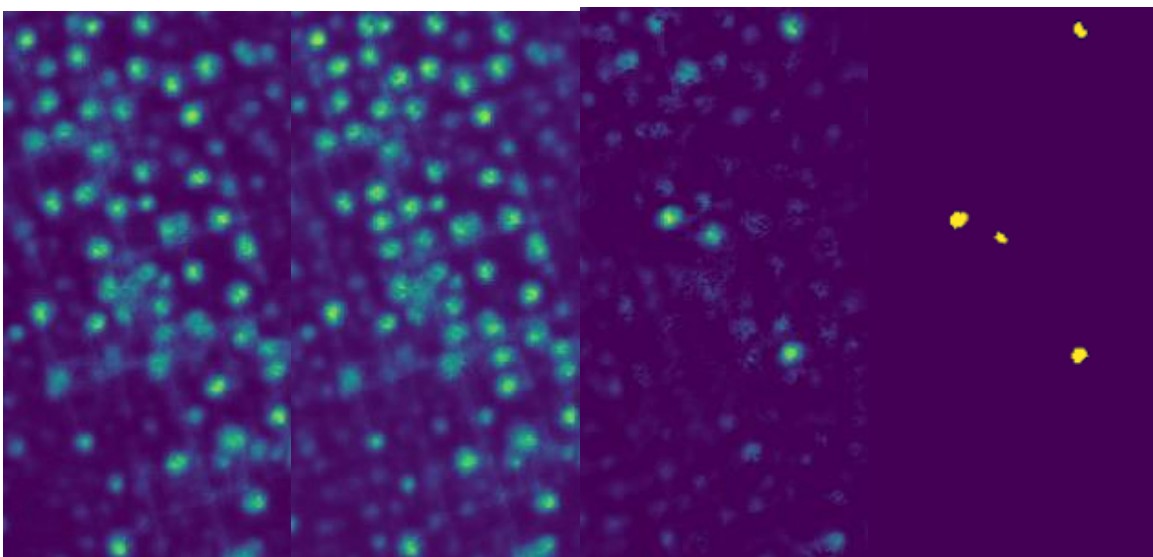

**Figure C3.** Before (top-left) and after (top-right) medium-resolution imagery (ESA, 2020). Same area in model output

(bottom, left to right): 1. before, 2. after, 3. difference, 4. detected new well pads.





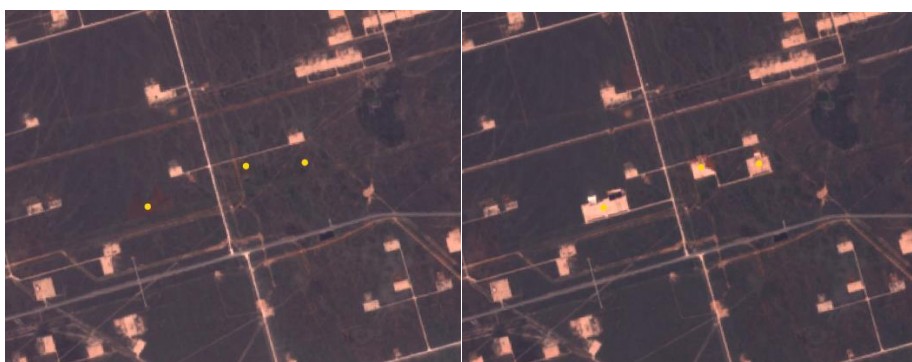

**Figure C4.** Example of an area where new development was found, before (top) and after (bottom) shown in Sentinel-2 imagery (ESA, 2020). Points in yellow indicate the locations of new well pad development.

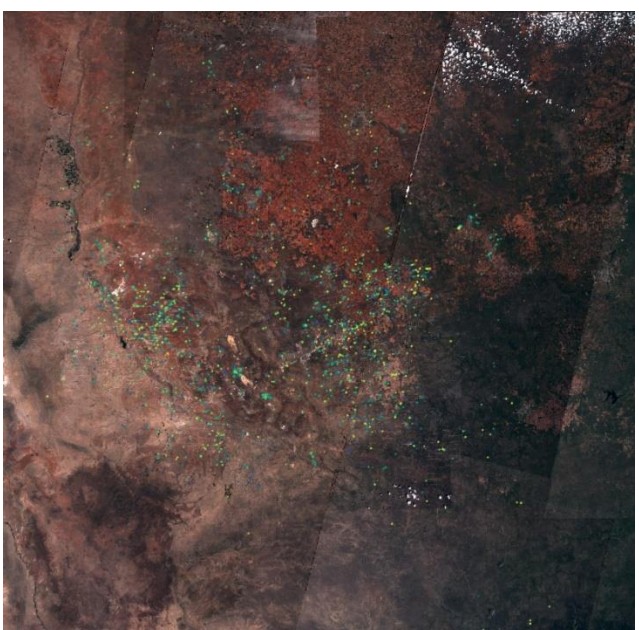

**Figure C5.** Number of new well pads constructed per month between August 1, 2019 and July 1, 2020 overlaid on Sentinel-2
imagery over a subset of the Permian Basin (ESA, 2020).



## Appendix D. Well completion emission estimates

Well completion flowback refers to the unconventional well development period following hydraulic fracturing in which water,
proppant, and entrained natural gas flow out of the wellbore to prepare a well for production (Allen et al., 2013). As of 2015,
U.S. federal regulations require all oil and gas wells except exploratory and low-pressure wells to utilize reduced emission
completions (RECs), which separate the natural gas and send to a pipeline as soon as technically feasible (USEPA, 2019);
occasionally, flaring or a combination of REC and flaring is used to partially control emissions. Previous research has
demonstrated that RECs control flowback emissions by an average of 99% (Allen et al., 2013). To estimate monthly
completion-related methane emissions within our 100 km x 100 km study area during the study period, we compiled a list of
every well located within our study area with a completion date between January 1 and April 30, 2020 (Enverus, 2020) and
applied two approaches to estimate potential and actual emissions. The first approach estimated actual emissions by applying
an emission factor (total methane emitted per well completion) based on 2018 data from 3,359 completions in the Permian
Basin reported to the EPA Greenhouse Gas Reporting Program, which operators estimate with a choice of measurements or
engineering equations (USEPA, 2019, 2020b). To convert total emissions into an hourly emission rate, we assumed that
completions emit at a constant rate over 4 days, the average duration from Allen et al. (2013). The second approach, which
estimated potential emissions, assumes that wells emit their initial gas production for 4 days following the completion date;
we assumed 80% methane content of natural gas and used the daily average production rate from the first complete month of
gas production (referred to as PracIP by (Enverus, 2020)).


The number of well completions per month in the study area dropped from 134 in January to 53 in April 2020. Based on
our first approach, January and April 2020 completion-related actual emissions were 2.5 and 1 Mg $CH_4$ $h^{-1}$, respectively, with
an average emission factor of 19 kg $CH_4$ $h^{-1}$ per completion and 93% of completions utilizing a REC or REC plus flaring
(Table D1). Based on the second approach, the average potential emission rate per completion was 61 kg $CH_4$ $h^{-1}$ in January
and 23 kg $CH_4$ $h^{-1}$ in April 2020; this results in total study area completion-related emissions of 9.3 and 1.9 Mg $CH_4$ $h^{-1}$ in
January and April, respectively (Table D2).





**Table D1**. Estimate of Permian Basin well completion emission factors based on US EPA Greenhouse Gas Reporting Program data.

| | Permian Basin Annual Completions (#) | Total Methane Emissions (Mg $CH_4$) | Average Emissions (Mg $CH_4$ completion$^{-1}$) | Average Emission Rate assuming 4 day duration (kg $CH_4$ completion$^{-1}$) |
|---|---|---|---|---|
| Reduced Emission Completion (REC) | 1,162 | 376 | 0.3 | 3 |
| REC & Flared | 1,955 | 4,673 | 2.4 | 25 |
| Uncontrolled | 14 | 35 | 2.5 | 26 |
| Flared | 228 | 1,202 | 5.3 | 55 |
| *Total* | *3,359* | *6,287* | *1.9* | *19* |





**Table D2.** Estimate of average monthly potential completion-related emissions from our study area from January 2019 – April 2020 based on initial production data and the assumption of 4 day completion duration.

| Year | Month | Average Ongoing Daily Well Completions (wells) | Average Aggregate Completion-Related Emissions (Mg $CH_4$ $h^{-1}$) |
|---|---|---|---|
| 2019 | 1 | 435 | 7.1 |
| 2019 | 2 | 616 | 15.2 |
| 2019 | 3 | 570 | 12.3 |
| 2019 | 4 | 706 | 12.0 |
| 2019 | 5 | 595 | 11.0 |
| 2019 | 6 | 569 | 9.3 |
| 2019 | 7 | 762 | 14.7 |
| 2019 | 8 | 884 | 13.6 |
| 2019 | 9 | 492 | 12.1 |
| 2019 | 10 | 658 | 10.5 |
| 2019 | 11 | 720 | 14.6 |
| 2019 | 12 | 461 | 9.4 |
| 2020 | 1 | 505 | 9.3 |
| 2020 | 2 | 335 | 7.7 |
| 2020 | 3 | 259 | 3.7 |
| 2020 | 4 | 212 | 1.9 |





**Appendix E. Oil & Gas production data and assessment of database completeness**

Production quantities of oil and gas from individual wells is reported to public state databases (RRC, 2020;NMOCD, 2020); however, the best results are achieved by analyses from an external database (Enverus, 2020) which filters and aggregates all of the publicly available datasets from all reporting agencies. Oil and Gas production data from New Mexico is updated on a monthly cadence, while data from Texas is updated twice each month but still only at monthly resolution. Timeseries of Oil,

Gas and combined barrels-of-oil equivalent (BOE) production within the greater Permian basin and 100 x 100 km study area are presented in Fig. E1. Similarly, Fig. E2 presents a timeseries of the number wells reporting production each month within the basin and 100 x 100km study area as well as timeseries of the number of wells exhibiting their first month of Oil and Gas production and their as their spud date: the date at which the subsurface drilling commences within the process of well development. The typical lag in data reporting is at least 3 months (Enverus, 2020) (e.g. O&G production during the month

of June is available on or shortly after the $1^{st}$ of September); however in practice reporting delays upwards of 6 months have been observed. We anticipate additional delays in the reporting of production data related to the global COVID-19 pandemic, thus here we attempt to broadly assess the incompleteness of the production dataset and its related impact on our estimates of the study area $CH_4$ loss rate.

The number of active wells reporting production was relatively constant in the Permian basin was relatively consistent through March 2020, only exhibiting a drop from the trend in April 2020 suggesting that new wells were coming online at roughly the same rate of older, depreciated wells being shut in. Alternatively, in the smaller 100 x 100 km study area which represents 7.4 $\pm$ 0.3 % ($1\sigma$) of the total Permian basin active well count for January 2019 – March 2020, the number of wells reporting production each month was increasing at a rate of $102 \pm 58$ ($1\sigma$) wells per month between January 2019 and March 2020.

During the same time span in the study area, the rate of new well production ($168 \pm 27$ wells/month, $1\sigma$) significantly outpaced the rate of depreciated wells being shut in by roughly a factor of 3.

Therefore, to estimate the complete dataset of total monthly production in the April-June 2020 under the timeframe of the observations of $CH_4$ emissions presented in Fig. 2, we extrapolate the average well count for January to March 2020 to the

subsequent three months as the dotted line on the Orange and Red traces of Fig. E2. We assume the deficit in wells reported represent the same distribution of oil and gas reported from each well present in the database; therefore, we linearly scale the production upwards by this factor as shown in the dotted lines in Fig. E1. This assessment suggests that that production largely plateaued during the height of the COVID-19 pandemic, rather than the <10% decrease observed by the reported data at time of submission. Therefore, using the projected gas production estimates, we calculate a projected loss rate in the basin from

both the monthly mean tower data and May aerial measurements the as the purple dotted line and yellow points respectively in Fig. E3.



This approach discussed above likely overestimates the oil and gas production due to the reduced activity observed from satellite well pad detection (Fig.7) and the reduced rate of new well development (Fig. E2). Therefore, we consider this to be

an upper limit on the study area gas production and therefore a lower limit on the $CH_4$ loss rate, with the actual value likely falling between the two estimates. Regardless, the adjusted loss rate represents a minimal adjustment within the 95% CI estimate expressed by the aerial and tower data temporal and analytical uncertainty that we do not consider it to differ significantly from the reported result in Fig. 6.





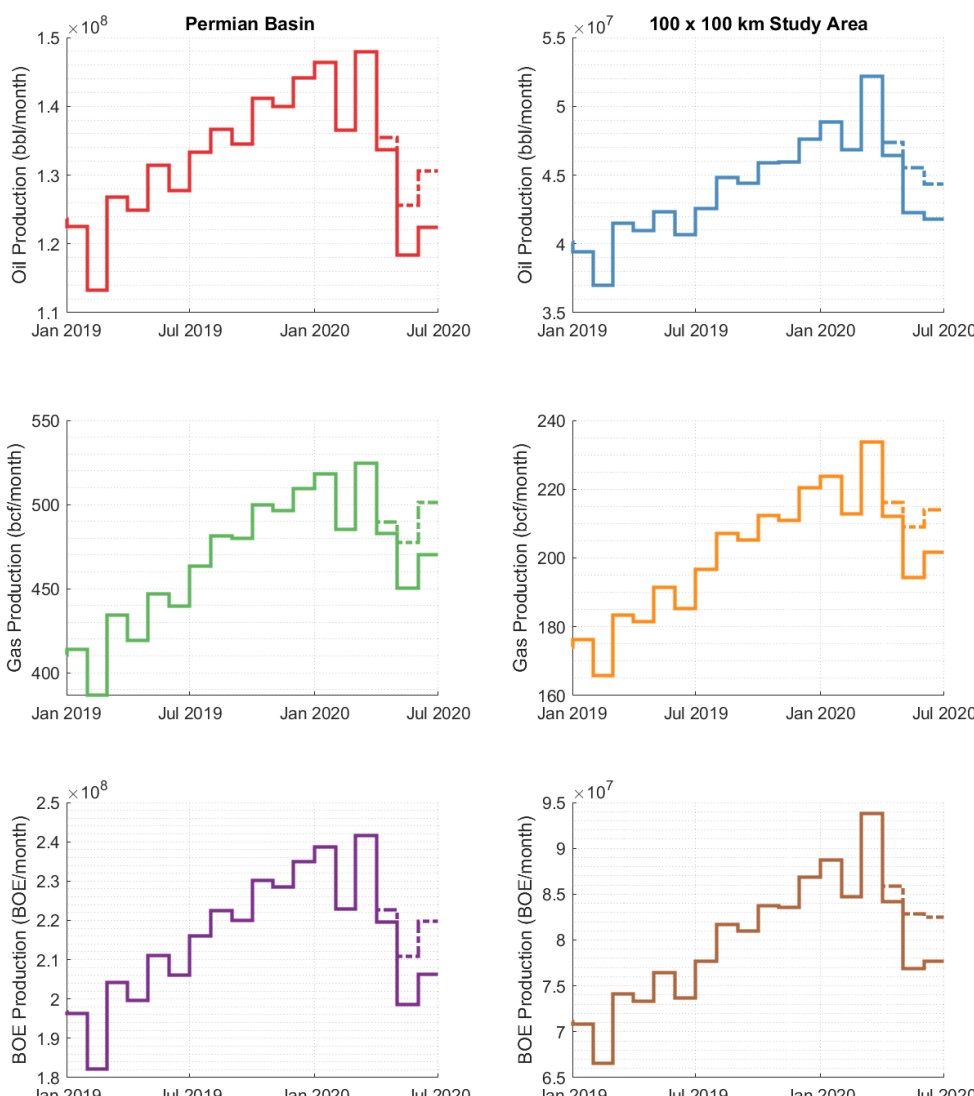


**Figure E1.** Monthly Timeseries of monthly Oil, Gas and Barrels of Oil Equivalent production (Top, Middle and Bottom respectively) in both the Permian Basin (left) and 100 x 100 km Study Area (Right). Dotted lines for April-June 2020 represent adjusted production to assess incompleteness of the dataset.






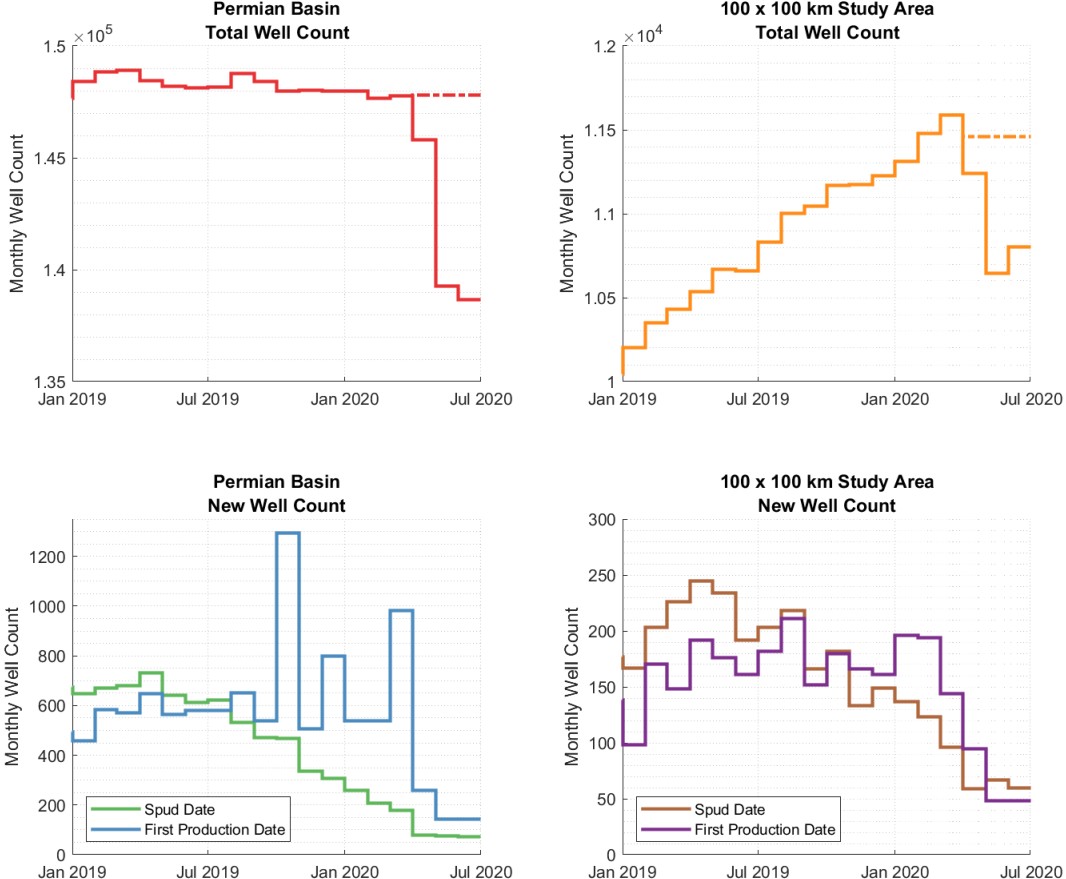

**Figure E2.** Monthly Timeseries of Active wells (top) and newly produced wells by spud date and month of first production (bottom) in both the Permian Basin (left) and 100 x 100 km Study Area (right). Dotted lines for April-June 2020 represent adjusted well counts to assess incompleteness of the dataset






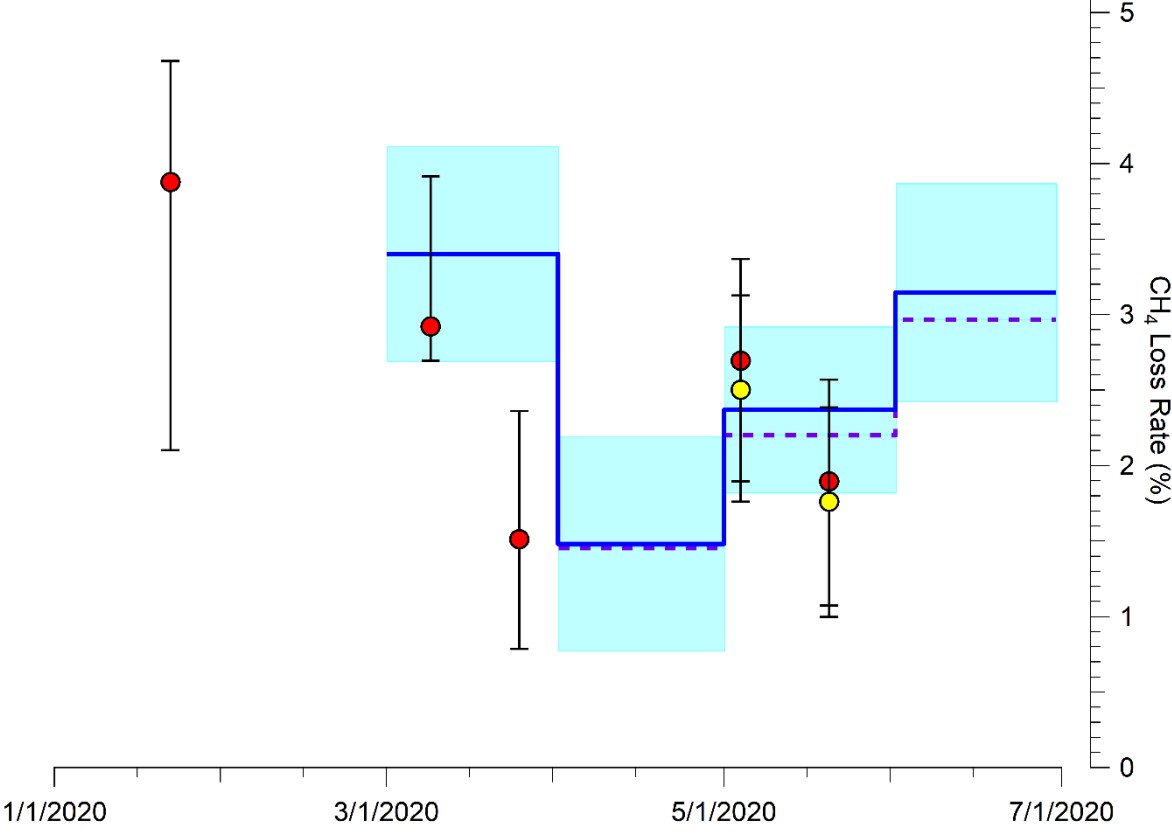

**Figure E3.** Gas production normalized loss rate in the 100 x 100 km study area following the same formatting as presented in Figure 6. Dotted purple line and yellow points reflect the adjusted tower and aerial based loss rates respectively after considering the incompleteness of the production database.




**Appendix F. Supplementary data from Bakken Shale**

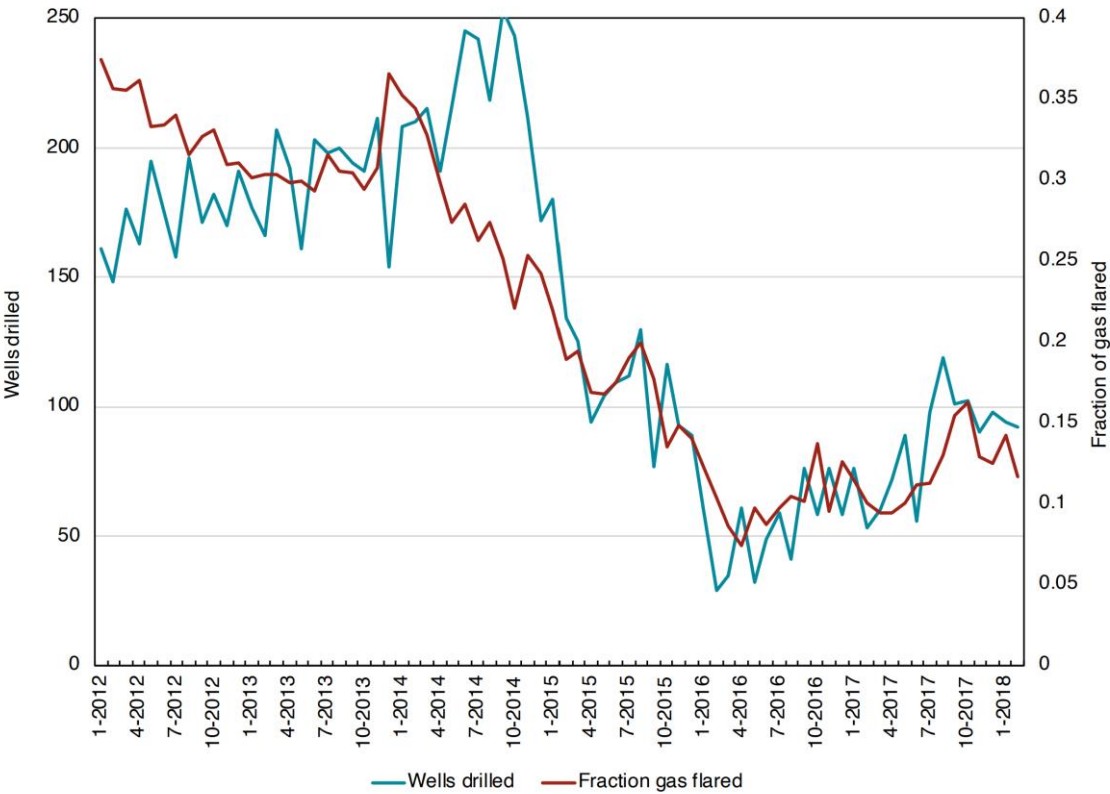

**Figure F1.** Number of wells drilled versus fraction of total gas production flared in the Bakken region (North Dakota, U.S.A.) from 2012 - 2017. Similar to trends observed in the Permian, there was a strong correlation between wells drilled and fraction of gas flared with both values decreasing rapidly when oil prices crashed in 2014.



**Data Availability**

Data are available for download at https://www.permianmap.org/

**Author Contribution**

D.R.L., B.H., A.R.B., M.K., E.A.K, A.J.M., and S.P.H. contributed to study conceptualization. D.R.L., B.H., R.G., M.O., K.R., Z.R.B., K.J.D., N.L.M, V.C.M., S.J.R., S.C., M.L.S., D.J.J, L.S., D.V.V., A.D., X.R., N.S., and K.T.S. contributed to methods development and data analysis. D.R.L., B.H., M.O., Z.R.B., K.J.D., M.L.S., D.J.V., and K.T.S. wrote the original

draft and all authors reviewed and edited the manuscript

**Competing Interests**

A.R.B., E.A.K., M.K., and A.J.M. serve on the PermianMAP scientific advisory panel. Authors declare no other competing interests.

**Acknowledgements**

The authors thank Beth Trask, Colin Leyden, Jon Goldstein, Louise White, Caleb Berman, and the entire PermianMAP team for their contribution to the project. We are grateful to Ramon Alvarez, Maureen Lackner, Ricardo Esparza, Ilse Aben, and Johannes Maasakkers for providing comments. We are grateful to Niall Armstrong for piloting the research aircraft and Leak Surveys, Inc. for performing the flare performance surveys. The PermianMAP project, which includes the aerial, tower, and flare survey data, is funded by Environmental Defense Fund; the work at Harvard University was supported by NASA Carbon

Monitoring System. Computations for this research were performed on The Pennsylvania State University's Institute for Computational and Data Sciences' Roar supercomputer.





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
