# Peer review of "Concurrent variation in oil and gas methane emissions and oil price during the COVID-19 pandemic"

_Atmospheric Chemistry and Physics, 2020_

## Referee Comment (RC1) · Anonymous Referee #1 · 11 Jan 2021

General remarks This is an interesting paper, and likely a precursor of a heavily populated suite of studies to come, looking at the many profound impacts of the pandemic. The paper strives to show that methane emissions from the US Permian Basin are closely linked to the oil price and hence to major economic factors such as the impact of Covid.

I have a general comment and some very minor specific notes listed below. The general comment is that the paper links emissions to oil price, not to gas price. Yes, I agree that gas emissions from oil production are important, but surely the obvious first link will be gas price rather than oil price? There is a gnomic remark in the abstract:

[Figure]

"a state of overcapacity in which...production exceeds midstream capacity and leads to high methane emissions". This then leads to the statement in L 407 "consequence of associated gas production increasing at a faster rate than midstream infrastructure capacity, which leads to extensive flaring and anomalous conditions related to excess gas throughput (e.g. pressure relief venting)." I'm not wholly clear here. Does that mean that more oil is being produced (because the oil price has gone up) and because the associated gas cannot go down the over-stuffed pipe, it is then vented? OK, might be so, but that hypothesis appears out of nowhere....Maybe explain this a bit more?

The paper has been rapidly done, but the findings appear soundly based and very interesting indeed. Publish with minor revisions.

Specific Comments

Page 2 Line 35 maybe an extra line to explain further this huge discrepancy? P2 L38. Paragraph break before 'The Permian Basin...? P5 Fig 2 caption needs to give the sources of data for this plot. P6 L115 "used 'the' Weather...." P7 L134 and also L136. Delaware sub-basin? P7 L143 the emissions magnitude.....are not. Trivial comment but the subject - 'magnitude' - is singular! Maybe say magnitudeS....(apologies – too much zoom frizzles the brain) P9 L183 – likewise – timeS ....are adjusted P10 L215 – dramatic rechange??? What does that mean? P10 L224 – remove 5 days - this is a bit obscure: I'm not sure what is meant here. Explain further? Is this simply excluding 3-sigma outliers? What's the impact of leaving them in? – in gasfield leaks the outliers can be significant. P14 L274 and also L288 – is there a connection between weather (i.e. clouds) and emissions? Is the prevalence of cloudy days in any way linked to the amount of gas pumped (and vented)? – or is the market so far away that distant gas demand and local weather are wholly disconnected. I'd assume most demand is from electric power needs (and heating) in Texas and nearby states? Is that correct? P21 L368-370. "Pandemic-related oil price crash" – yes, but this remark needs to be buttressed by a reference or other factual proof. A casual concurrence is not necessarily proof of a causal connection. Also this is oil price. All this discussion

has its focus on oil price, not gas. . ..How about Henry Hub gas price?- surely that is the parameter to postulate as the controlling factor, not oil price? – although I'd agree that contractual gas prices tend to be ratio-ed to oil price. P20 and 21 "G&P" – acronym soup – I know this is defined in L 378 but why not say 'gathering and processing'. It's not much longer and saves a lot of misery. O&G is another, especially as the paper conflates gas price into oil price.

---

## Referee Comment (RC2) · Anonymous Referee #2 · 22 Feb 2021

This manuscript describes a combination of different atmospheric measurements— airborne and tower-based with inversion analysis— to estimate changes in methane emissions from oil and gas production in the Permian Basin over a period of large fluctuations in the price of oil. The authors report a large decrease (more than a factor of 3) in methane emissions from the study region coincident with a threefold reduction in oil prices comparing Jan-mid Mar 2020 with late Mar through the end of April. The analysis convincingly documents a change in emissions through several different lines of evidence, and is a laudable step forward in process understanding of methane emissions from oil and gas production in North America.

[Figure]

Overall, the study is clear, but some improvements to organization could be made to improve communication with the readers of ACP. I recognize that the authors are presenting many different datasets from many different techniques, which is challenging to do. Currently there is too much mixing of methods and results, and the order of things appears to be a bit off. Most of the data (results, I would say, including the most important figures/tables) were presented in the methods section (section 2) rather that in the results section (section 3). I would recommend the authors carefully separate the methods and results sections, and put all data (findings: figures, tables, etc.) in the results. It would be helpful for the authors to use sub-heads in the results section to help guide the flow of data, including results from the main atmospheric analyses and the supporting information about flares and well starts.

It was also a bit unusual that there was no discussion section, instead this section was titled "conclusions". There was some mixing of methods even into the results section. Line 342 is the first time VIIRS data is mentioned, and this should probably be at least mentioned in the methods. Upon re-reading, I saw that it was mentioned in lines 111-112, but a bit more context in the first description would be helpful. Methods to describe results presented in Figure 7 need to be fleshed out more as well in the methods section preferably. Some figures were presented out of sequence (e.g. Figure 4). A revision of the text with some attention to readability and a consistent order of data presentation is recommended.

Minor comments:

Incorrectly capitalized letters randomly scattered throughout (e.g., Figure 1 caption: lowercase "black" when referring to a color, Emissions on line 320)

Lines 146-7: "Our assumption that emissions are proportional to gas production should provide a reasonable estimate of the spatial pattern of emissions corresponding to well locations." Can you provide a citation or further information for this reasoning?

Line 201: "16Z (11 LST) through 22Z (17 LST)" Don't know what Z means in this

case, and please spell out that LST=local standard time (I assume) Line 208: Figure 4 presented out of sequence (prior to fig. 3) Line 209: "Although though" Line 215: what is "rechange"? Line 268: please provide a citation for this sentence: "Here we consider only higher-quality XCH4 measurements (quality assurance value > 0.5)." Don't know what a quality assurance value is. Figure 3: how were aerial & tower-based measurements combined to get 1 estimate? Apologies if I missed this.

---

## Author Comment (AC1) · 7 Mar 2021

Comments from referees/public

Author responses are indented below specific comments. Page and line numbers refer to the original draft.

Anonymous Referee 1

General remarks This is an interesting paper, and likely a precursor of a heavily populated suite of studies to come, looking at the many profound impacts of the pandemic. The paper strives to show that methane emissions from the US Permian Basin are

closely linked to the oil price and hence to major economic factors such as the impact of Covid.

I have a general comment and some very minor specific notes listed below. The general comment is that the paper links emissions to oil price, not to gas price. Yes, I agree that gas emissions from oil production are important, but surely the obvious first link will be gas price rather than oil price? There is a gnomic remark in the abstract: "a state of overcapacity in which production exceeds midstream capacity and leads to high methane emissions". This then leads to the statement in L 407 "consequence of associated gas production increasing at a faster rate than midstream infrastructure capacity, which leads to extensive flaring and anomalous conditions related to excess gas throughput (e.g. pressure relief venting)." I'm not wholly clear here. Does that mean that more oil is being produced (because the oil price has gone up) and because the associated gas cannot go down the over-stuffed pipe, it is then vented? OK, might be so, but that hypothesis appears out of nowhere. Maybe explain this a bit more? The paper has been rapidly done, but the findings appear soundly based and very interesting indeed. Publish with minor revisions. We thank the reviewer for their comments. In the introduction (P2 L60), we have added the statement "In the Permian Basin, oil price is a stronger driver of well development than natural gas price." For our concluding hypothesis (P22 L407), we have expanded the sentence to better explain the causal relationship of oil price and methane emissions. Page 2 Line 35 maybe an extra line to explain further this huge discrepancy?

P2 L35: We added an explanation of the discrepancy between top-down and bottom-up approaches to the end of the sentence: "...primarily due to abnormal emissions that are difficult to quantify with bottom-up approaches".

P2 L38. Paragraph break before 'The Permian Basin?

P2 L38: paragraph break added

P5 Fig 2 caption needs to give the sources of data for this plot.

P5 Figure 2: source added (Baker Hughes, 2020)

P6 L115 "used 'the' Weather."

P6 L115: "the" added before WRF-CHEM model

P7 L134 and also L136. Delaware sub-basin?

Changed "Delaware Basin" to "Delaware sub-basin" throughout manuscript

P7 L143 the emissions magnitude are not. Trivial comment but the subject - 'magnitude' - is singular! Maybe say magnitudeS: : :.(apologies – too much zoom frizzles the brain)

P7 L143: changed "are" to "is"

P9 L183 – likewise – timeS are adjusted

P9 L183: changed "are" to "is"

P10 L215 – dramatic rechange??? What does that mean?

P10 L215: replaced "rechange in behavior" with "decline in"

P10 L224 – remove 5 days – this is a bit obscure: I'm not sure what is meant here. Explain further? Is this simply excluding 3-sigma outliers? What's the impact of leaving them in? – in gasfield leaks the outliers can be significant.

After checking the data, we determined that 4 days were removed due to methane enhancements that exceeded the modeled concentration by 3 sigma or greater. In all four cases, the enhancement was 10x higher than expected due to unknown circumstances, such as a very large, nearby upwind source. The outlier days have emission estimates in the range of 500 – 600 Mg/hr, 3 – 9x higher than average emissions, but comprise less than 4

P14 L274 and also L288 – is there a connection between weather (i.e. clouds) and emissions? Is the prevalence of cloudy days in any way linked to the amount of gas

pumped (and vented)? – or is the market so far away that distant gas demand and local weather are wholly disconnected. I'd assume most demand is from electric power needs (and heating) in Texas and nearby states? Is that correct?

We do not expect there is a relationship between regional cloud cover and methane emissions in the Permian Basin. There are several markets that purchase oil and gas from the Permian including petrochemical facilities, so economic metrics such as oil price should have much greater impacts than local demand. Although clouds should not affect emission rates, it is possible that some emission detection approaches would be less effective on cloudy days, which potentially could result in less effective leak mitigation. However, this likely has minimal impact on study area emissions since most operators only perform leak detection at most a few times per year. Our study also utilizes two approaches: the towers and aircraft, that can quantify emissions on cloudy days. P21 L368-370. "Pandemic-related oil price crash" – yes, but this remark needs to be buttressed by a reference or other factual proof. A casual concurrence is not necessarily proof of a causal connection. Also this is oil price. All this discussion has its focus on oil price, not gas. How about Henry Hub gas price?- surely that is the parameter to postulate as the controlling factor, not oil price? – although I'd agree that contractual gas prices tend to be ratio-ed to oil price.

P21 L368-370: changed "pandemic-related" to "pandemic-associated". In the introduction (P2 L57), we cite a source (Reed and Krauss, 2020) about the likely causal relationship between the pandemic, lower economic activity, decreased fuel consumption, and lower oil prices. As discussed in response to the first comment, Permian basin development rates are controlled more by oil price than natural gas price.

P20 and 21 "GP" – acronym soup – I know this is defined in L 378 but why not say 'gathering and processing'. It's not much longer and saves a lot of misery. OG is another, especially as the paper conflates gas price into oil price.

We have removed the acronym "GP" and replaced throughout with "gathering and pro-

cessing" or "midstream". We believe OG is a commonly used acronym and have retained it when used to describe the oil and natural gas supply chain or companies, but replaced "OG production" with "oil and natural gas production" when directly referring to production data.

Anonymous Referee 2

This manuscript describes a combination of different atmospheric measurements airborne and tower-based with inversion analysis to estimate changes in methane emissions from oil and gas production in the Permian Basin over a period of large fluctuations in the price of oil. The authors report a large decrease (more than a factor of 3) in methane emissions from the study region coincident with a threefold reduction in oil prices comparing Jan-mid Mar 2020 with late Mar through the end of April. The analysis convincingly documents a change in emissions through several different lines of evidence, and is a laudable step forward in process understanding of methane emissions from oil and gas production in North America.

Overall, the study is clear, but some improvements to organization could be made to improve communication with the readers of ACP. I recognize that the authors are presenting many different datasets from many different techniques, which is challenging to do. Currently there is too much mixing of methods and results, and the order of things appears to be a bit off. Most of the data (results, I would say, including the most important figures/tables) were presented in the methods section (section 2) rather that in the results section (section 3). I would recommend the authors carefully separate the methods and results sections, and put all data (findings: figures, tables, etc.) in the results. It would be helpful for the authors to use sub-heads in the results section to help guide the flow of data, including results from the main atmospheric analyses and the supporting information about flares and well starts.

It was also a bit unusual that there was no discussion section, instead this section was titled "conclusions". There was some mixing of methods even into the results

section. Line 342 is the first time VIIRS data is mentioned, and this should probably be at least mentioned in the methods. Upon re-reading, I saw that it was mentioned in lines 111-112, but a bit more context in the first description would be helpful. Methods to describe results presented in Figure 7 need to be fleshed out more as well in the methods section preferably. Some figures were presented out of sequence (e.g. Figure 4). A revision of the text with some attention to readability and a consistent order of data presentation is recommended.

We thank the reviewer for their comments. We have reorganized the paper so that all the tower and TROPOMI results (and corresponding figures and tables) are first presented in the results section. We add a description of VIIRS when the data are first mentioned in line 111. The caption of Figure 7 has been expanded to describe its relationship to Appendix 2. The sequence of figures and tables have been corrected to match the order they first referred to in the texts. We have changed the heading "Conclusions" to "Discussion".

Minor comments:

Incorrectly capitalized letters randomly scattered throughout (e.g., Figure 1 caption: lowercase "black" when referring to a color, Emissions on line 320)

We have corrected improperly capitalized letters.

Lines 146-7: "Our assumption that emissions are proportional to gas production should provide a reasonable estimate of the spatial pattern of emissions corresponding to well locations." Can you provide a citation or further information for this reasoning?

P7 L155: We have changed "corresponding to well locations" to "corresponding to the location of oil and natural gas infrastructure" to clarify and added a reference (Maasakkers et al., 2016)

Line 201: "16Z (11 LST) through 22Z (17 LST)" Don't know what Z means in this case, and please spell out that LST=local standard time (I assume)

P9 L209: We have spelled out "local standard time" and replace Z with "UTC"

Line 208: Figure 4 presented out of sequence (prior to fig. 3)

Switched numbers of Figures 3 and 4

Line 209: "Although though"

P9 L209: deleted "though"

Line 215: what is "rechange"?

Typo and awkward wording. As discussed in response to referee 1, we replaced "rechange in behavior" with "decline in"

Line 268: please provide a citation for this sentence: "Here we consider only higher-quality XCH4 measurements (quality assurance value > 0.5)." Don't know what a quality assurance value is.

P11 268: We have added the reference describing how quality assurance values are calculated for TROPOMI XCH4 data (Apituley et al., 2017)

Figure 3: how were aerial tower-based measurements combined to get 1 estimate? Apologies if I missed this.

For the monthly averages, we average all accepted aerial and tower-based estimates with equal weights. To clarify, we edited the caption of Figure 4 (previously Figure 3) to add "which weights tower and aircraft-based estimates equally."

Author changes in manuscript

In addition to the changes outlined above, we have made several updates to the manuscript by incorporating updated production and activity data. Although we did not extend the emission time series featured in the paper, there was a small change in tower-based methane emission estimates ( 10 Mg/hr increase) due to updated quality assurance metrics that remove outliers based new data past this time period. The

updated production data changes the loss rate results from 3.4

Figures 2, 3, 4, E1, E2, and Tables 1 and D2 have been updated with more recent data.

Appendix D has been updated with new well completion data (June – August 2020).

Appendix E has been updated to reflect the addition of new oil and natural gas production data.

Please also note the supplement to this comment:
https://acp.copernicus.org/preprints/acp-2020-1175/acp-2020-1175-AC1-supplement.pdf

**Supplement:**

*Concurrent variation in oil and gas methane emissions and oil price during the COVID-19 pandemic*

https://acp.copernicus.org/preprints/acp-2020-1175/

**Comments from referees/public**

> *Author responses are indented and italicized below specific comments. Page and line numbers refer to the original*

5 > *draft.*

**Anonymous Referee #1**

General remarks This is an interesting paper, and likely a precursor of a heavily populated suite of studies to come, looking at the many profound impacts of the pandemic. The paper strives to show that methane emissions from the US Permian Basin are closely linked to the oil price and hence to major economic factors such as the impact of Covid.

10 I have a general comment and some very minor specific notes listed below. The general comment is that the paper links emissions to oil price, not to gas price. Yes, I agree that gas emissions from oil production are important, but surely the obvious first link will be gas price rather than oil price? There is a gnomic remark in the abstract: "a state of overcapacity in which production exceeds midstream capacity and leads to high methane emissions". This then leads to the statement in L 407 "consequence of associated gas production increasing at a faster rate than midstream infrastructure capacity, which leads to

15 extensive flaring and anomalous conditions related to excess gas throughput (e.g. pressure relief venting)." I'm not wholly clear here. Does that mean that more oil is being produced (because the oil price has gone up) and because the associated gas cannot go down the over-stuffed pipe, it is then vented? OK, might be so, but that hypothesis appears out of nowhere. Maybe explain this a bit more? The paper has been rapidly done, but the findings appear soundly based and very interesting indeed. Publish with minor revisions.

> *We thank the reviewer for their comments. In the introduction (P2 L60), we have added the statement "In the Permian*
>
> *Basin, oil price is a stronger driver of well development than natural gas price." For our concluding hypothesis (P22*
>
> *L407), we have expanded the sentence to better explain the causal relationship of oil price and methane emissions.*

25 Page 2 Line 35 maybe an extra line to explain further this huge discrepancy?

> *P2 L35: We added an explanation of the discrepancy between top-down and bottom-up approaches to the end of the*
> *sentence: "...primarily due to abnormal emissions that are difficult to quantify with bottom-up approaches".*

30 P2 L38. Paragraph break before 'The Permian Basin?

> *P2 L38: paragraph break added*

P5 Fig 2 caption needs to give the sources of data for this plot.

> *P5 Figure 2: source added (Baker Hughes, 2020)*

P6 L115 "used 'the' Weather."

40 > *P6 L115: "the" added before WRF-CHEM model*

P7 L134 and also L136. Delaware sub-basin?

> *Changed "Delaware Basin" to "Delaware sub-basin" throughout manuscript*

P7 L143 the emissions magnitude are not. Trivial comment but the subject - 'magnitude' - is singular! Maybe say magnitudeS: : :.(apologies – too much zoom frizzles the brain)

*P7 L143: changed "are" to "is"*

P9 L183 – likewise – timeS are adjusted

*P9 L183: changed "are" to "is"*

P10 L215 – dramatic rechange??? What does that mean?

*P10 L215: replaced "rechange in behavior" with "decline in"*

P10 L224 – remove 5 days – this is a bit obscure: I'm not sure what is meant here. Explain further? Is this simply excluding 3-sigma outliers? What's the impact of leaving them in? – in gasfield leaks the outliers can be significant.

After checking the data, we determined that 4 days were removed due to methane enhancements that exceeded the modeled concentration by 3 sigma or greater. In all four cases, the enhancement was ~10x higher than expected due to unknown circumstances, such as a very large, nearby upwind source. The outlier days have emission estimates in the range of 500 – 600 Mg/hr, ~3 – 9x higher than average emissions, but comprise less than 4% of the data so have minor impact on the overall results.

P14 L274 and also L288 – is there a connection between weather (i.e. clouds) and emissions? Is the prevalence of cloudy days in any way linked to the amount of gas pumped (and vented)? – or is the market so far away that distant gas demand and local weather are wholly disconnected. I'd assume most demand is from electric power needs (and heating) in Texas and nearby states? Is that correct?

*We do not expect there is a relationship between regional cloud cover and methane emissions in the Permian Basin. There are several markets that purchase oil and gas from the Permian including petrochemical facilities, so economic metrics such as oil price should have much greater impacts than local demand. Although clouds should not affect emission rates, it is possible that some emission detection approaches would be less effective on cloudy days, which potentially could result in less effective leak mitigation. However, this likely has minimal impact on study area emissions since most operators only perform leak detection at most a few times per year. Our study also utilizes two approaches: the towers and aircraft, that can quantify emissions on cloudy days.*

P21 L368-370. "Pandemic-related oil price crash" – yes, but this remark needs to be buttressed by a reference or other factual proof. A casual concurrence is not necessarily proof of a causal connection. Also this is oil price. All this discussion has its focus on oil price, not gas. How about Henry Hub gas price?- surely that is the parameter to postulate as the controlling factor, not oil price? – although I'd agree that contractual gas prices tend to be ratio-ed to oil price.

*P21 L368-370: changed "pandemic-related" to "pandemic-associated". In the introduction (P2 L57), we cite a source (Reed and Krauss, 2020) about the likely causal relationship between the pandemic, lower economic activity, decreased fuel consumption, and lower oil prices. As discussed in response to the first comment, Permian basin development rates are controlled more by oil price than natural gas price.*

P20 and 21 "G&P" – acronym soup – I know this is defined in L 378 but why not say 'gathering and processing'. It's not much longer and saves a lot of misery. O&G is another, especially as the paper conflates gas price into oil price.

*We have removed the acronym "G&P" and replaced throughout with "gathering and processing" or "midstream". We believe O&G is a commonly used acronym and have retained it when used to describe the oil and natural gas supply chain or companies, but replaced "O&G production" with "oil and natural gas production" when directly referring to production data.*

**Anonymous Referee #2**

This manuscript describes a combination of different atmospheric measurements airborne and tower-based with inversion analysis to estimate changes in methane emissions from oil and gas production in the Permian Basin over a period of large fluctuations in the price of oil. The authors report a large decrease (more than a factor of 3) in methane emissions from the study region coincident with a threefold reduction in oil prices comparing Jan-mid Mar 2020 with late Mar through the end of April.

The analysis convincingly documents a change in emissions through several different lines of evidence, and is a laudable step forward in process understanding of methane emissions from oil and gas production in North America.

Overall, the study is clear, but some improvements to organization could be made to improve communication with the readers of ACP. I recognize that the authors are presenting many different datasets from many different techniques, which is challenging to do. Currently there is too much mixing of methods and results, and the order of things appears to be a bit off. Most of the data (results, I would say, including the most important figures/tables) were presented in the methods section (section 2) rather that in the results section (section 3). I would recommend the authors carefully separate the methods and results sections, and put all data (findings: figures, tables, etc.) in the results. It would be helpful for the authors to use sub-heads in the results section to help guide the flow of data, including results from the main atmospheric analyses and the supporting information about flares and well starts.

It was also a bit unusual that there was no discussion section, instead this section was titled "conclusions". There was some mixing of methods even into the results section. Line 342 is the first time VIIRS data is mentioned, and this should probably be at least mentioned in the methods. Upon re-reading, I saw that it was mentioned in lines 111-112, but a bit more context in the first description would be helpful. Methods to describe results presented in Figure 7 need to be fleshed out more as well in the methods section preferably. Some figures were presented out of sequence (e.g. Figure 4). A revision of the text with some attention to readability and a consistent order of data presentation is recommended.

*We thank the reviewer for their comments. We have reorganized the paper so that all the tower and TROPOMI results (and corresponding figures and tables) are first presented in the results section. We add a description of VIIRS when the data are first mentioned in line 111. The caption of Figure 7 has been expanded to describe its relationship to Appendix 2. The sequence of figures and tables have been corrected to match the order they first referred to in the texts. We have changed the heading "Conclusions" to "Discussion".*

Minor comments:

Incorrectly capitalized letters randomly scattered throughout (e.g., Figure 1 caption: lowercase "black" when referring to a color, Emissions on line 320)

*We have corrected improperly capitalized letters.*

Lines 146-7: "Our assumption that emissions are proportional to gas production should provide a reasonable estimate of the spatial pattern of emissions corresponding to well locations." Can you provide a citation or further information for this reasoning?

*P7 L155: We have changed "corresponding to well locations" to "corresponding to the location of oil and natural gas infrastructure" to clarify and added a reference (Maasakkers et al., 2016)*

Line 201: "16Z (11 LST) through 22Z (17 LST)" Don't know what Z means in this case, and please spell out that LST=local standard time (I assume)

150        *P9 L209: We have spelled out "local standard time" and replace Z with "UTC"*

Line 208: Figure 4 presented out of sequence (prior to fig. 3)

           *Switched numbers of Figures 3 and 4*
155
Line 209: "Although though"

           P9 L209: d*eleted "though"*

160  Line 215: what is "rechange"?

           *Typo and awkward wording. As discussed in response to referee 1, we replaced "rechange in behavior" with "decline in"*

165  Line 268: please provide a citation for this sentence: "Here we consider only higher-quality XCH4 measurements (quality assurance value > 0.5)." Don't know what a quality assurance value is.

           *P11 268 and References: We have added the reference describing how quality assurance values are calculated for TROPOMI XCH4 data (Apituley et al., 2017)*
170
Figure 3: how were aerial & tower-based measurements combined to get 1 estimate? Apologies if I missed this.

           *For the monthly averages, we average all accepted aerial and tower-based estimates with equal weights. To clarify, we edited the caption of Figure 4 (previously Figure 3) to add "which weights tower and aircraft-based estimates*
175        *equally."*

**Author changes in manuscript**

In addition to the changes outlined above, we have made several updates to the manuscript by incorporating updated production
180  and activity data. Although we did not extend the emission time series featured in the paper, there was a small change in tower-based methane emission estimates (~10 Mg/hr increase) due to updated quality assurance metrics that remove outliers based new data past this time period. The updated production data changes the loss rate results from 3.4% to 3.3% (pre-crash) and 1.5% to 1.9% (minimum).

185  Figures 2, 3, 4, E1, E2, and Tables 1 and D2 have been updated with more recent data.

Appendix D has been updated with new well completion data (June – August 2020).

Appendix E has been updated to reflect the addition of new oil and natural gas production data.

[revised manuscript text omitted]